# REM sleep is associated with distinct global cortical dynamics and controlled by occipital cortex

Ziyue Wang[1,2,6], Xiang Fei[1,3,6], Xiaotong Liu[1,3,6], Yanjie Wang[1,2,6], Yue Hu[1,4], Wanling Peng[1], Ying-wei Wang[4], Siyu Zhang ✉[2] ✉ & Min Xu ✉[1,5] ✉

The cerebral cortex is spontaneously active during sleep, yet it is unclear how this global cortical activity is spatiotemporally organized, and whether such activity not only reflects sleep states but also contributes to sleep state switching. Here we report that cortex-wide calcium imaging in mice revealed distinct sleep stage-dependent spatiotemporal patterns of global cortical activity, and modulation of such patterns could regulate sleep state switching. In particular, elevated activation in the occipital cortical regions (including the retrosplenial cortex and visual areas) became dominant during rapid-eye-movement (REM) sleep. Furthermore, such pontogeniculooccipital (PGO) wave-like activity was associated with transitions to REM sleep, and optogenetic inhibition of occipital activity strongly promoted deep sleep by suppressing the NREM-to-REM transition. Thus, whereas subcortical networks are critical for initiating and maintaining sleep and wakefulness states, distinct global cortical activity also plays an active role in controlling sleep states.

The brain comprises a highly interconnected, complex neural network. The overall dynamic changes of neural activity within this network are collectively referred to as "brain states". Wakefulness and sleep are distinct brain states that, in mammals, are mainly defined by activity patterns of the thalamocortical system, as reflected by electroencephalogram (EEG) and motor activity[1,2]. The different sleep stages are also observed in non-mammalian species[3–5]. Distinct cortical activity associated with various sleep-wake stages is generally considered a reflection of each sleep state and plays a limited role in sleep state switching[6–9], although emerging evidence supports an active role for the cortex in controlling sleep as well[10,11]. However, from the perspective of network theory, the activity of key nodes within a complex network may have a broad impact on various properties of the network[12]. Therefore, by providing many critical nodes of the brain network, the cerebral cortex may play an active role in regulating the

dynamic change of brain states, including the transition between different sleep stages.

On the other hand, it is increasingly recognized that even simple behaviors may require coordinating neural activity in multiple brain regions[13–15]. Consistent with this notion, sleep control has also been shown to involve a large number of brain regions that are distributed primarily in subcortical areas[6–9]. It is thus important to obtain a global view of how ongoing activity in various brain regions contributes to the regulation and function of sleep. Recent developments in large-scale, high-speed recording, and manipulation of global neural dynamics at the mesoscale allow us to dissect the role of various cortical regions. In this study, we used "global neural activity" to denote the large-scale neural activity across large areas in the brain, for example, the entire dorsal part of the cortex or the whole brain.

[1]Institute of Neuroscience, State Key Laboratory of Neuroscience, Center for Excellence in Brain Science and Intelligence Technology, Chinese Academy of Sciences, 200031 Shanghai, China. [2]Collaborative Innovation Center for Brain Science, Department of Anatomy and Physiology, Shanghai Jiao Tong University School of Medicine, 200025 Shanghai, China. [3]University of Chinese Academy of Sciences, 100049 Beijing, China. [4]Department of Anesthesiology, Huashan Hospital, Fudan University, 200040 Shanghai, China. [5]Shanghai Center for Brain Science and Brain-Inspired Intelligence Technology, 201210 Shanghai, China. [6]These authors contributed equally: Ziyue Wang, Xiang Fei, Xiaotong Liu, Yanjie Wang. ✉e-mail: zhang_siyu@sjtu.edu.cn; mxu@ion.ac.cn

It is now generally appreciated that the brain is highly active during sleep[16]. There are characteristic differences between global neural activity patterns during sleep and wakefulness, as revealed by EEG recording or functional magnetic resonance imaging (fMRI) recording in mammals, although with relatively low resolution[2,17,18], or by cellular resolution imaging in non-mammalian species[4,19]. However, due to the considerable differences in the brain networks of different evolved animals, it is necessary to measure the global network dynamics of the mammalian brain during different sleep states with a high spatiotemporal resolution, and determine how these activity control sleep.

In the current study, we used mesoscale Ca²⁺ imaging to examine the global neural activity feature in sleep with a high spatiotemporal resolution from the entire dorsal cortex of mice, and uncovered an occipital activity pattern that is essential for promoting the transition from NREM to REM sleep. Our results demonstrate that dynamic cortical activity could play an active role in regulating sleep states.

## Results

### Mesoscale Ca²⁺ imaging of cortical activity during sleep

To determine whether there are specific global activity patterns that associate with different sleep states, we recorded the macroscopic

neuronal activity dynamics with a high spatiotemporal resolution using widefield-of-view optical imaging in the mice. We measured Ca²⁺ activity from the entire dorsal part of the cortex from *Thy1*-GCaMP6s mice[20] using a transparent skull preparation (mice with intact but optically transparent skulls)[21,22], when the mice were trained to sleep under the microscope (Fig. 1a and Supplementary Movie 1). The *Thy1*-GCaMP6s mice express a genetically encoded Ca²⁺ indicator, GCaMP6s[23], driven by the *Thy1* promoter, providing stable access to pyramidal neurons' activity in cortical layers 2/3 and layer 5[20,24]. The current method allows the recording of Ca²⁺ activity from the motor, somatosensory and visual cortices, and some association areas[21,22,24] (Fig. 1b, c). On the other hand, activity measured with mesoscale Ca²⁺ imaging reflects the summation of Ca²⁺ signals from a large number of neurons, so it may not be equivalent to electrophysiologically recorded population spiking activity. In this study, we used "cortical activity" to specifically refer to the Ca²⁺ signals obtained with widefield imaging.

During imaging, mice were head-fixed under the microscope with body and paws moving freely, and no external stimulus was applied. All imaging experiments were performed during the day with the lights on. EEG (from the left auditory cortex) and EMG (from the neck muscle) were recorded to determine the sleep-wake states of the mice. To

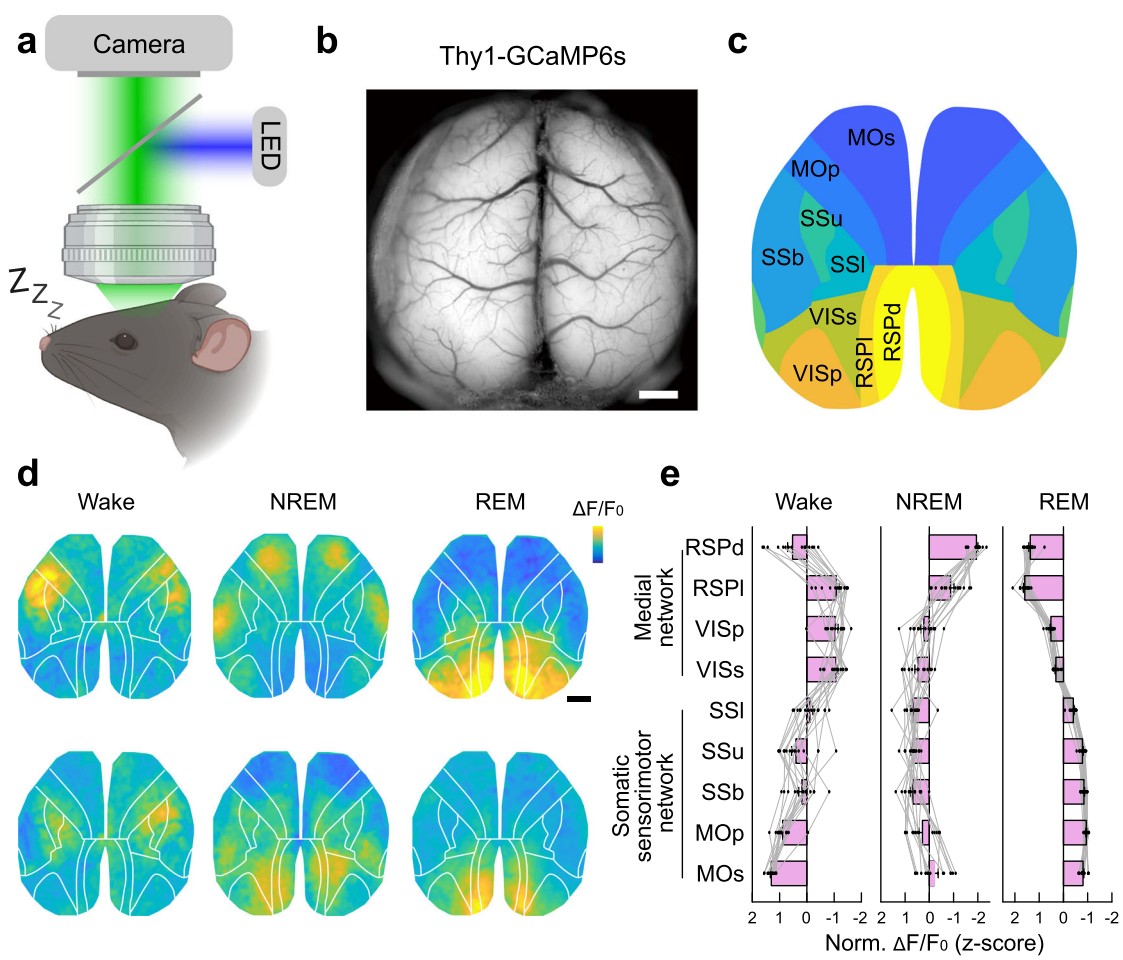

**Fig. 1 | Mesoscale Ca²⁺ imaging of the dorsal cortex during the sleep-wake cycle.** **a** Schematic diagram depicting the mesoscale Ca²⁺ imaging from head-fixed mice. **b** Example field-of-view (time-averaged) of the dorsal cortex imaged through the transparent skull of a Thy1-GCaMP6s mouse. Scale, 1 mm. **c** Cortex atlas for alignment. MOs secondary motor cortex, MOp primary motor cortex, SSb somatosensory cortex, barrel field, SSu somatosensory cortex, upper limb, SSl somatosensory cortex, lower limb, VISp primary visual cortex, VISs association visual cortex, RSPd

dorsal retrosplenial cortex, RSPl lateral retrosplenial cortex. **d** Representative Ca²⁺ activity during different brain states. Scale (ΔF/F₀, z-score): Wake, −1.2 to 1.2; NREM, −1.2 to 1.2; REM, −2 to 2. Black bar, 1 mm. **e** Normalized activation across the recorded brain regions in different brain states. The averaged activity in each brain state was z-score normalized across the nine brain regions; thus, positive values mean more activation. n = 15 sessions from 5 mice. Data are mean ± SEM. Raw data for **e** are provided in a Source Data file.

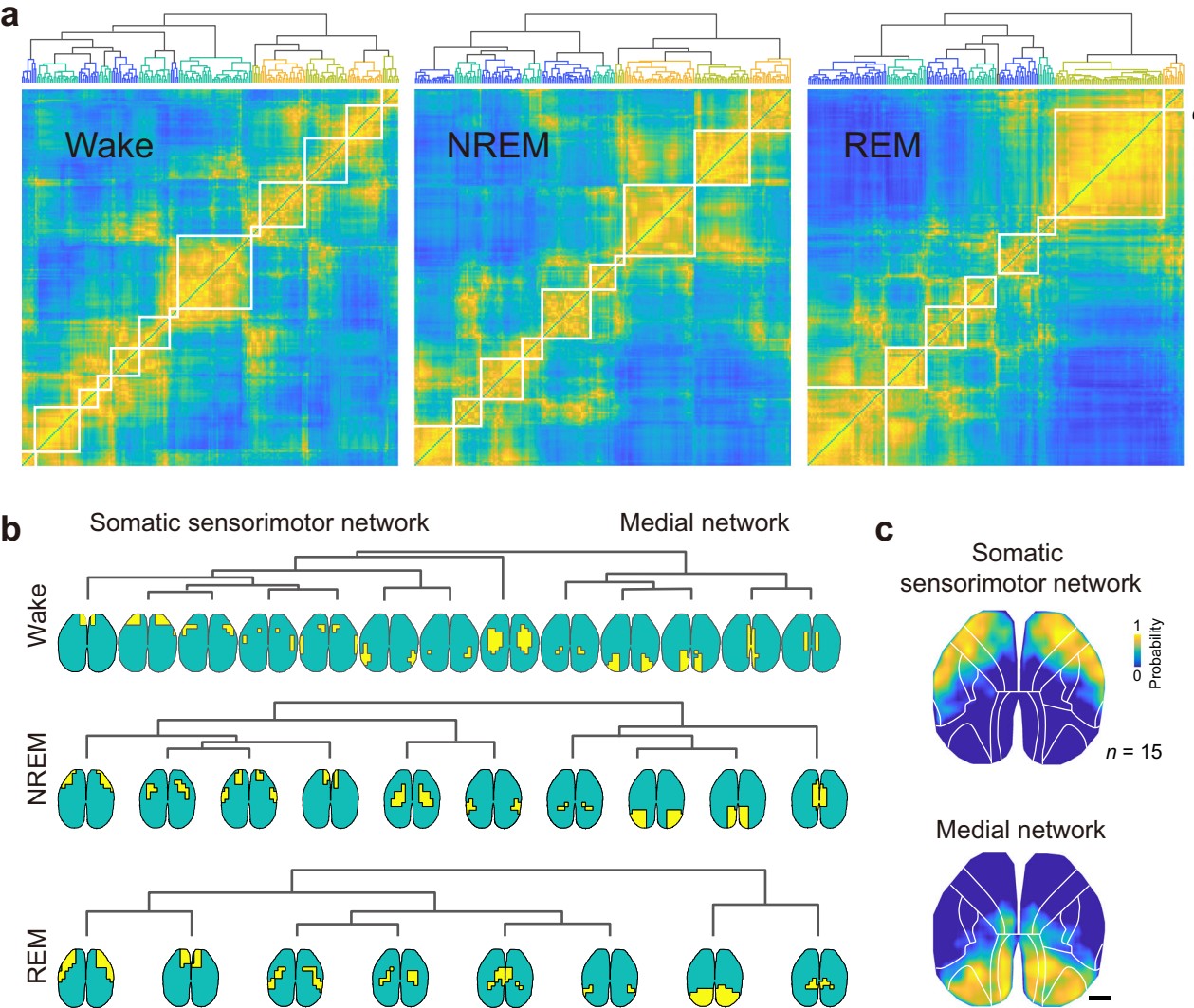

**Fig. 2 | Anterior-posterior organization of cortical activity during sleep.**
**a** Regional activity correlation matrix during Wake (left), NREM (middle), and REM (right) in one example recording (-70 min). The pairwise correlation was computed using neural activity during Wake, NREM, or REM in each small patch (-10 ×10 pixels) of the brain region. Brain regions with similar activity profiles were grouped together by applying a hierarchical clustering analysis. Scale (cc), −1 to 1. **b** Cortical networks revealed by the hierarchical clustering analysis. The dendrogram shows each grouped brain region (highlighted in yellow) with similar activity. **c** Regions of the somatic sensorimotor network and the medial network. Shown are regions of the two networks averaged from 15 recordings. Scale, 1 mm.

score the brain states, we performed fast Fourier transforms (FFTs) on EEG to extract the δ (0.5−4 Hz) and θ (6−10 Hz) activity. Brain states were classified according to established criteria:[25,26] Wakefulness, desynchronized EEG and high EMG activity; NREM, synchronized EEG with high δ activity and low EMG activity; REM, high EEG θ power and low EMG activity. The habituated mice spent the majority of time in sleep (NREM: 65%, Wake: 27%, REM: 8%; Supplementary Fig. 1), with diverse facial movements and occasionally gross body movements during wakefulness.

Various movements profoundly shape cortical activity, producing distinct patterns (Fig. 1d and Supplementary Movie 2). For example, we found that the facial movements (extracted from the face camera) were significantly associated with elevated Ca²⁺ activity in the somatosensory cortex and motor cortex during wakefulness (Supplementary Fig. 2). On the other hand, we also observed highly structured activity patterns during sleep (Fig. 1d and Supplementary Movie 2), suggesting region-specific dynamic neuronal activations in sleep[27]. To be noted, consistent with previous reports[28], the head-fixed mice in our recording typically slept with their eyes open, and the pupil size varied in different sleep states (Supplementary Fig. 3). The change in

pupil size can have significant effects on cortical activity, especially in the visual cortex. However, multiple lines of our evidence suggest that changes in pupil size did not significantly contribute to the cortical activity observed in our recording: First, although both the pupil size and activity in the primary visual cortex (V1) showed oscillatory dynamics during NREM sleep, there was no positive correlation between these two signals; instead, they tended to be negatively correlated (Pearson's $r = −0.41 \pm 0.041$, mean ± SEM; Supplementary Fig. 3a, b), with a phase difference close to zero ($0.50 \pm 0.85$ s, mean ± SEM; Supplementary Fig. 3c). This negative correlation suggests that both cortical activity and pupil reflect different aspects of brain state oscillations during NREM sleep. Second, during REM sleep, the pupil remains mostly constricted;[28] however, V1 activity often showed large transient increases (Supplementary Fig. 3d). The two signals during REM were also negatively correlated (Pearson's $r = −0.41 \pm 0.044$, mean ± SEM; Supplementary Fig. 3d, e). These results support that cortical activity during sleep is mainly spontaneous, unlike during wakefulness.

We began by analyzing the activity of each brain region during different sleep-wake states, and found three types of modulation that

were strongly associated with functions and anatomical proximity of different brain regions (Supplementary Fig. 4): The motor cortex was more active during wakefulness and suppressed during REM sleep, in contrast to the visual cortex and the retrosplenial cortex (RSP) in the medial cortical network[29], which showed an opposing modulation pattern that was more active during REM sleep; The somatosensory cortex exhibited more activation during NREM sleep. Additionally, active wakefulness (with more facial movements) and quiet wakefulness were also associated with different cortical patterns (Supplementary Fig. 5).

Besides the region-specific modulation in discrete cortical regions, macroscopic imaging allows for identifying global activity patterns. We thus normalized activity in each brain state across the entire dorsal cortex and determined the relative activation of each cortical area. Consistent with the above result, the most active region during wakefulness was the motor cortex, while the most active region during REM sleep was the RSP; The NREM activity had a relatively uniform distribution across the dorsal cortex, except that the RSP showed the least activity (Fig. 1e and Supplementary Movie 2). This brain region-specific modulation suggests that different cortical regions may be specifically involved in regulating different sleep-wake states or implementing state-specific functions in the sleep-wake cycle.

### Highly structured cortical activation patterns during sleep

The above analysis used time-averaged activity, which reflected the relative activation probability of each brain region during different brain states. Also, brain atlas-based quantification can be affected by the accuracy of atlas-matching procedures. We therefore examined cortical activity patterns in different sleep-wake states on a fine time scale and without prior compartment of the cortex.

We first performed a functional connectivity analysis[30,31], in which we computed the similarity of neural activity dynamics between small grids of cortical areas and used hierarchical clustering to group regions with similar activity. We found that the cortex could be divided into different subnetworks using activity either from NREM or REM sleep (Fig. 2a, b and Supplementary Fig. 6), and the clustering analysis revealed two distinct subnetworks: The first network consisted of a large part of the somatic sensorimotor network, and the second network mainly covered the medial cortical network (Fig. 2c). This result suggested that activity in the dorsal cortex during sleep was organized primarily according to the anterior-posterior axis, further supporting the functional divergence of these brain regions during different sleep states.

We next performed the principal component analysis (PCA) to detect the structured spatial patterns during the sleep-wake cycle. PCA reliably extracted highly similar activity patterns from different mice (Supplementary Fig. 7), similar to the previous analysis using seeds-based correlation[32]. Figure 3a shows the five most reliably detected principal components (PCs, PC1 – PC5), which together accounted for $17.7\% \pm 3.0\%$ (mean ± SEM) of the total variance ($n = 15$ recordings from 5 mice; Supplementary Fig. 8a). The major PCs were generally symmetric across the two hemispheres (Supplementary Fig. 8b) and highly correlated with anatomical modules of the cortex (Supplementary Fig. 8c). For example, PC1 mainly reflected activation of the visual cortex and RSP, and PC2 was primarily in the somatosensory cortex (Fig. 3a and Supplementary Fig. 8c). These results showed that the ongoing spontaneous activity was primarily organized according to intrinsic anatomical connectivity of the cortex[33], reflecting synchronized activation across the dorsal cortex during the sleep-wake cycle.

These PCs had distinct dynamics during the sleep-wake cycle (Fig. 3b), which were reflected in changes in PCA coefficients, quantitative measures of the correlation between each PC and cortical activity patterns during each imaging frame—higher coefficients mean higher similarity. PC1, PC4, and PC5 were more active during REM sleep and less active during wakefulness and NREM sleep ($P < 0.001$,

Wilcoxon sign-rank test; Fig. 3b, c); PC2 was more active during NREM sleep than during both wakefulness and REM sleep, in contrast to the opposing activity pattern of PC3 ($P < 0.001$, Wilcoxon sign-rank test). These modulations mirrored our previous analysis using atlas-based activity segmentation methods, further supporting the region-specific activity changes during different sleep-wake states. In addition, the PCs showed fast temporal dynamics, showing as transient activity events (Fig. 3d, e) (e.g., event duration: PC1, $0.92 \pm 0.02$ s; PC2, $0.90 \pm 0.03$ s; $n = 4$ mice; mean ± SEM).

The PCs exhibited prominent oscillations, similar to previous reports[30–32]. Spectral analysis revealed strong power in the δ band (0.5–4 Hz) and slow oscillations (<1 Hz; Fig. 3f, g). However, although these oscillations showed a brain state-dependent manner, with the highest oscillation power during REM sleep (Supplementary Fig. 9a), they did not correlate with EEG δ or slow waves (Pearson's r < 0.01, for all PCs) (Supplementary Fig. 9b, c).

### Distinct occipital activity pattern during REM sleep

We next focused our analysis on cortical activity patterns during REM sleep. PCA (using only data in REM sleep) showed that REM activity was dominated by the first PC of REM sleep (PC1^REM), which accounted for $27.5\% \pm 2.6\%$ (mean ± SEM) of the total variance, much larger than other PCs (explained variance by PC2–4: 1.1–3.3; $n = 15$ recordings from 5 mice; Supplementary Fig. 10a). We thus only analyzed the PC1^REM in all following sections.

The PC1^REM primarily reflected activation of the occipital cortical network, especially the RSP (Fig. 4a, Supplementary Fig. 10b), consistent with previous reports[34–36]. The PC1^REM showed a large dynamic change in its coefficient (Fig. 4b and Supplementary Fig. 11) and often appeared as transient events. Further analysis showed that this intermittent activation is associated with different sub-stages of REM sleep (tonic vs. phasic REM, or quiet vs. active REM)[37]—We defined phasic and tonic REM according to the presence of phasic facial movements (extracted from the face camera) and found that the PC1^REM was significantly higher ($P < 0.001$, Wilcoxon signed rank test; Supplementary Fig. 12) during phasic REM, suggesting different REM stages exhibit distinct cortical activation patterns. In addition, the increased activation of the PC1^REM also correlated with more general activation of the cortex, and it often occurred as a spreading activation pattern starting from the RSP (Fig. 4c, Supplementary Fig. 13, and Supplementary Movie 4 and 5).

We next further examined the dynamic change of cortical activation during REM sleep, particularly its relationship with other features associated with REM, such as EEG θ power and eye movements. The fluctuation of PC1^REM moderately correlated with the EEG power ratio between the θ and δ band (Fig. 4d, e and Supplementary Movie 3; Pearson's $r = 0.41 \pm 0.03$, $n = 4$ mice), consistent with the fact that they both reflected activation of the cortex[38]. The occurrence of burst eye movements, which were extracted from a video of the mouse pupil, was also associated with the increase of PC1^REM (Fig. 4d, f and Supplementary Movie 3); however, such correlation was not observed during wakefulness (Supplementary Fig. 14).

Another feature of brain activity during REM sleep is the PGO waves, which originate in the pons and propagate through the lateral geniculate nucleus to the occipital cortex[39]. The unique occipital activity we observed may represent the cortical activity of mouse PGO waves. We therefore examined this by recording local field potentials (LFP) from the pons during the mesoscale imaging (Fig. 4g). We identified the pontine waves (P-wave, the pontine part of the PGO waves) as large negative potential in the LFP[40,41], and we found that occipital activity significantly increased immediately after the P-wave (Fig. 4h, i; peak latency, $0.89 \pm 0.07$ s, mean ± SEM). This result support that mice also have PGO waves, and the structured occipital activity during REM sleep may represent the cortical activation of the PGO waves.

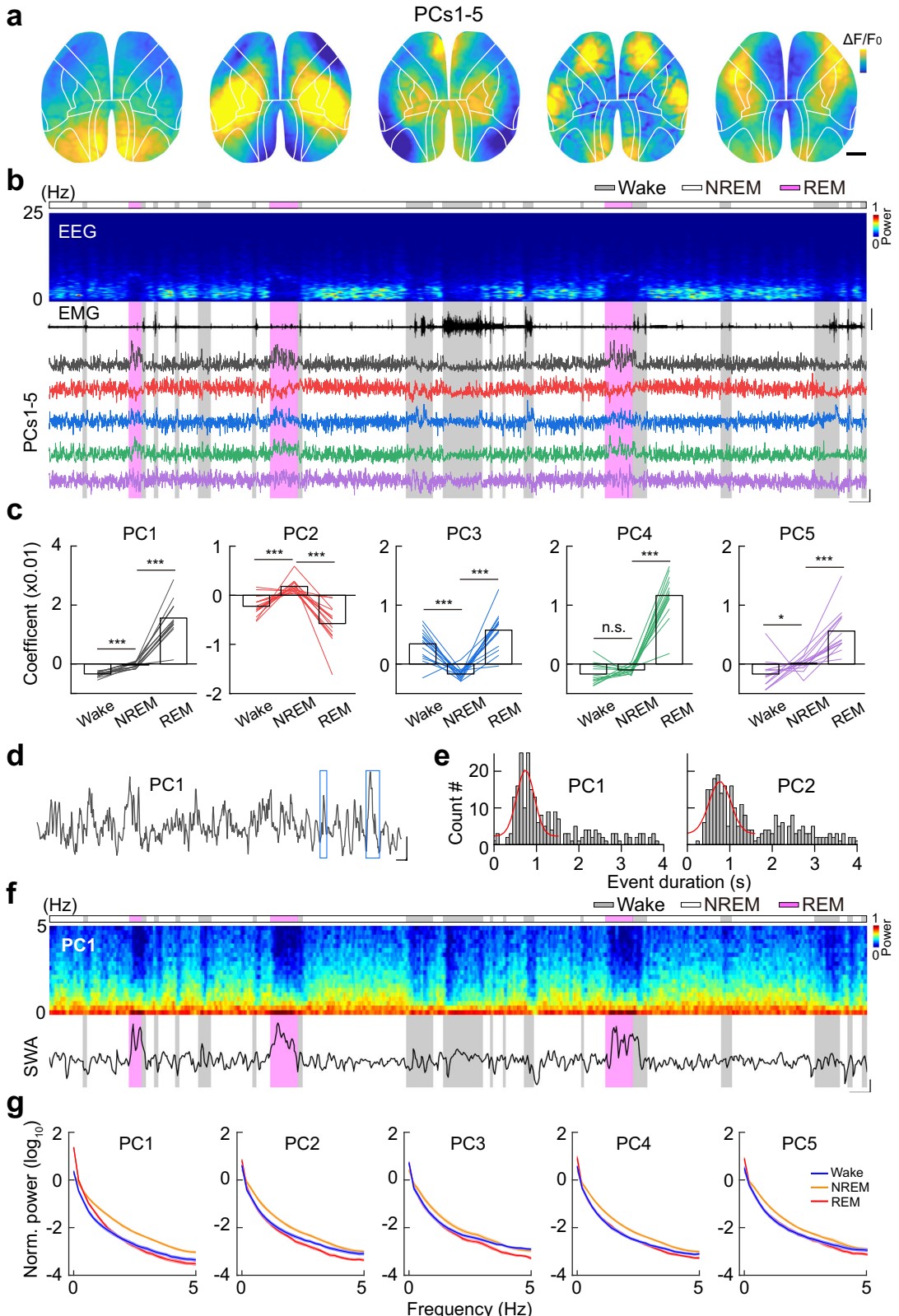

### Occipital activity dynamic during NREM sleep

Although the occipital cortex was highly active during REM sleep, we noticed that the PC1$^{REM}$-like activity pattern (REM-like pattern) also appeared in other brain states. We thus next analyzed the REM-like pattern during wakefulness and NREM by computing the similarity between the activity pattern of each imaging frame and the PC1$^{REM}$ (Fig. 5a). At the population level, there was no correlation between

PC1$^{REM}$ and the activity pattern in both wakefulness and NREM (Pearson's $r = -0.11 \pm 0.01$ and $-0.046 \pm 0.007$ for Wake and NREM, respectively; mean $\pm$ SEM; $P < 0.0001$, Paired $t$-test, all the $t$-test used the current study is two-tailed; $n = 15$ recording from 5 mice; Fig. 5b). However, while cortical patterns during wakefulness were generally not similar with the PC1$^{REM}$ (except during the drowsy states when the pupil was constricted and lack of facial movements. Supplementary

**Fig. 3 | Highly structured cortical activation patterns during the sleep-wake cycle. a** Activation patterns in an example recording extracted using PCA, showing the most frequently detected five major PCs during the sleep-wake cycle. Scale ($\Delta F/F_0$, z-score): PC1, −2 to 2; PCs 2–5, −1.2 to 1.2. Black bar, 1 mm. **b** Time course of EEG spectrum and the coefficient of PCs showing in panel **a**. Top to bottom, EEG power spectrogram, EMG (scale, 0.5 mV), coefficient of PCs (scale, 0.02 and 100 s). The brain states are color-coded; the same color code is used in all following figures. **c** Coefficient of the five major PCs in different brain states. The same color code was used for each PC, as shown in panel **b**. Each line represents data from one recording. *$P < 0.05$, ***$P < 0.001$; Two-tailed Student's paired $t$ test or Wilcoxon

signed-rank test. Data of PC1 - PC5 were from 15, 15, 15, 13, and 14 sessions from 5 mice, respectively. In this and all subsequent figures, summary data are expressed as the mean ± SEM. **d** An example trace showing the time course of the PC1 coefficient. The two blue boxes indicate two activity events. Scale: 2 s and 0.02. **e** Duration of the cortical activity events. $n = 235$ and 264 events for PC1 and PC2, respectively. **f** Time course of the PC1 spectrum and the normalized mean power (0–5 Hz) in a log scale. Time scale, 100 s. **g** Normalized power spectrum of the PCs. Shading is SEM. $n = 15$ recording from 5 mice. Raw data for **c** and exact $P$ values are provided in a Source Data file.

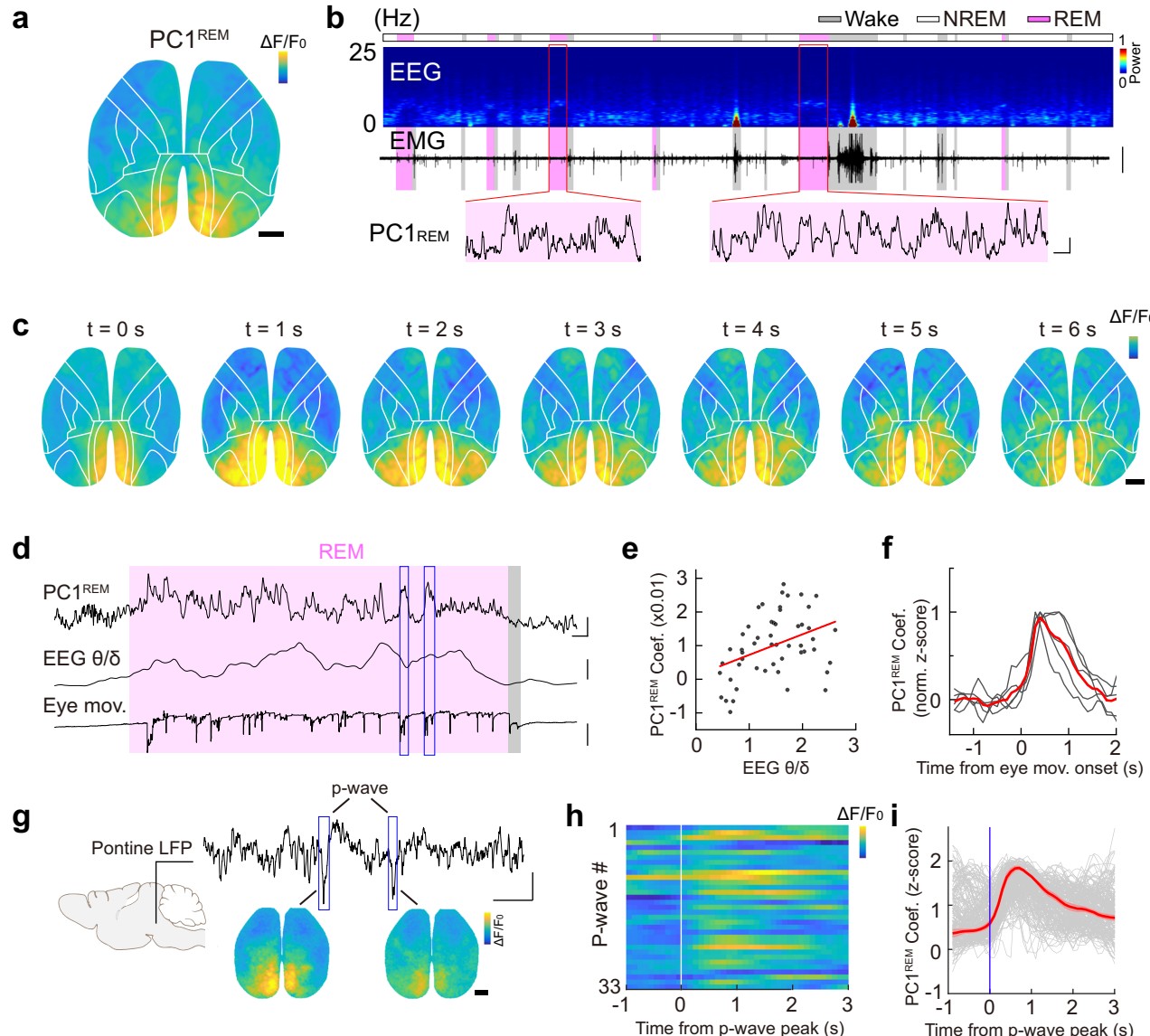

**Fig. 4 | Cortical activation patterns during REM sleep. a** The first PC of REM activity. Scale ($\Delta F/F_0$, z-score): −2 to 2. **b** Time course of PC1$^{REM}$, showing a large dynamic change of the PC. Scale: EMG, 0.5 mV; coefficient of PC, scale, 0.01 and 10 s. **c** An example showing the dynamic change of the spreading activity during REM sleep. Scale ($\Delta F/F_0$, z-score): −1.5 to 1.5. **d** An example showing the PC1$^{REM}$ coefficient, EEG power ratio between θ and δ band, and eye movements in the horizontal direction. Scale: PC1, 0.02 and 10 s; EEG ratio, 1; eye movements, 10 pixels. The two blue boxes indicate two bouts of eye movements. **e** Correlation between PC1$^{REM}$ coefficient and EEG power ratio between θ and δ band. **f** Correlation between eye movements and PC1$^{REM}$. PC1$^{REM}$ coefficient was aligned

using the onset of eye movement. Each black trace represents averaged change of PC1$^{REM}$ coefficient from one recording ($n = 9$–29 events). The red trace is the group average ($n = 4$ mice). **g** Left, schematic diagram depicting LFP recording from the pons. Right, Examples of P-waves (Scale, 0.1 mV and 1 s) and associated cortical activity (Scale, $\Delta F/F_0$, z-score: −1.5 to 1.5). **h–i** Occipital activity increased immediately after the P-waves. PC1$^{REM}$ coefficient aligned to the negative peak of p-waves detected from one recording (**h**, $n = 33$) or from all 5 mice (**i**, $n = 199$). Scale in **h**, $\Delta F/F_0$ (z-score): −1.5 to 1.5. All black bars in the figure represent 1 mm. Raw data for **e** are provided in a Source Data file.

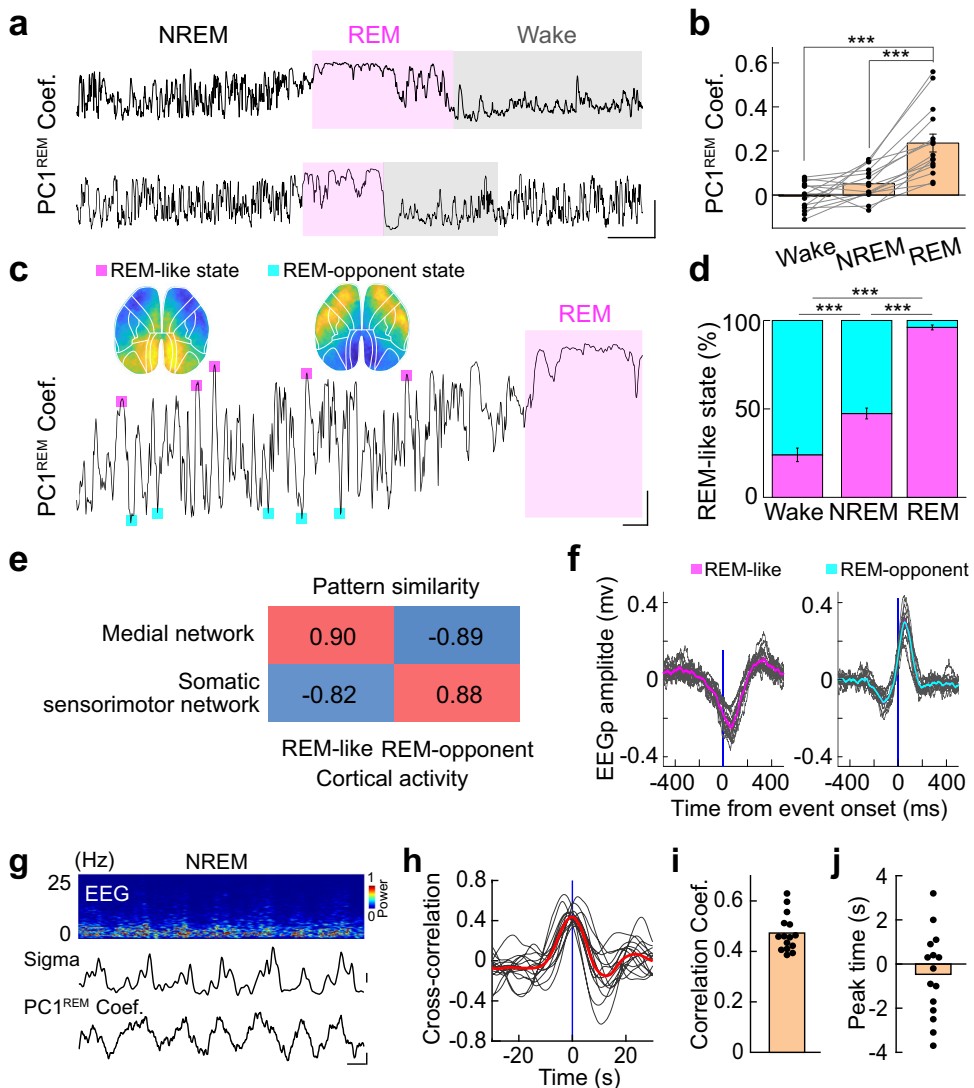

**Fig. 5 | Occipital activity dynamics during the sleep-wake cycle. a** Dynamic change of the PC1$^{REM}$ in different brain states from two example traces. Scale, 1 and 20 s. The black line is the Pearson correlation coefficient between each image frame and the PC1$^{REM}$. **b** Quantification of PC1$^{REM}$ dynamics in different brain states. $P < 0.0001$ for both Wake *vs.* REM and NREM *vs.* REM; Two-tailed paired $t$-test. $n = 15$ recordings from 5 mice. Data are mean ± SEM. **c** Dynamic change of the REM-like state during NREM. The magenta and cyan squares marked the REM-like events and REM-opponent events, respectively. The cortical activity patterns corresponding to the two states were shown above the correlation curve. Scale, 0.3 and 5 s. **d** Quantification of the REM-like state (magenta) and REM-opponent state (cyan) in different brain states. $P < 0.0001$ for all the comparisons; Two-tailed

paired $t$-test. $n = 15$ recordings from 5 mice. Data are mean ± SEM. **e** Cortical activity pattern during the REM-like state and REM-opponent state was tightly associated with the occipital network and anterolateral network, respectively. Shown is the Pearson correlation coefficient between each comparison. Data are mean ± SEM. **f** The average EEG waveform associated with REM-like events and REM-opponent events. $n = 15$ recordings from 5 mice. **g** Top to bottom, EEG power spectrogram, Sigma oscillations (scale, 0.5 a.u.), coefficient of PC1$^{REM}$ (scale, 0.02 and 100 s). **h–j** Cross-correlation analysis of EEG sigma oscillations and PC1$^{REM}$ coefficient. $n = 15$ recordings from 5 mice. Raw data for **b, d, i, j** and exact $P$ values are provided in a Source Data file.

Fig. 15), the NREM activity frequently exhibited a high positive or negative correlation with the PC1$^{REM}$ in a transient manner (Fig. 5c), suggesting that cortical activity patterns during NREM oscillate between a REM-like state and an opposing state (REM-opponent state). Further analysis indicated that the occurrence of REM-like and REM-opponent cortical activity patterns showed a strong brain state-dependent manner−REM sleep was dominated by REM-like patterns, and wakefulness had more REM-opponent patterns (REM-like *vs.* REM-opponent: REM, 23:1; Wake, 1:3.2; NREM, 1:1.1) (Fig. 5d).

The REM-like state showed high activity in the medial network and low activity in the somatic sensorimotor network, while the REM-opponent state had opposing patterns with more activation in the somatic sensorimotor network (Fig. 5c and Supplementary Fig. 16). Indeed, these two cortical patterns were highly similar to the occipital

network and anterolateral network revealed by our regional correlation analysis (Fig. 5e). Additionally, we found that the REM-like state and REM-opponent state were associated with different EEG waveforms that were measured from the temporal part of the cortex (Fig. 5f).

It has been recently found that EEG during NREM sleep exhibits infra-slow oscillations in the sigma band[42,43]. We next examined the relation between the sigma oscillations and the REM-like activity. We chose long NREM bouts with few micro-arousals and calculated EEG sigma power (Fig. 5g). Cross-correlation analysis revealed a significant positive correlation (Pearson's $r = 0.47 ± 0.02$; mean ± SEM) between the two signals (Fig. 5h, i), with no apparent difference in the peak time at the group level although the sigma oscillations may lead or lag the REM-like activity for a few seconds (Fig. 5h, j).

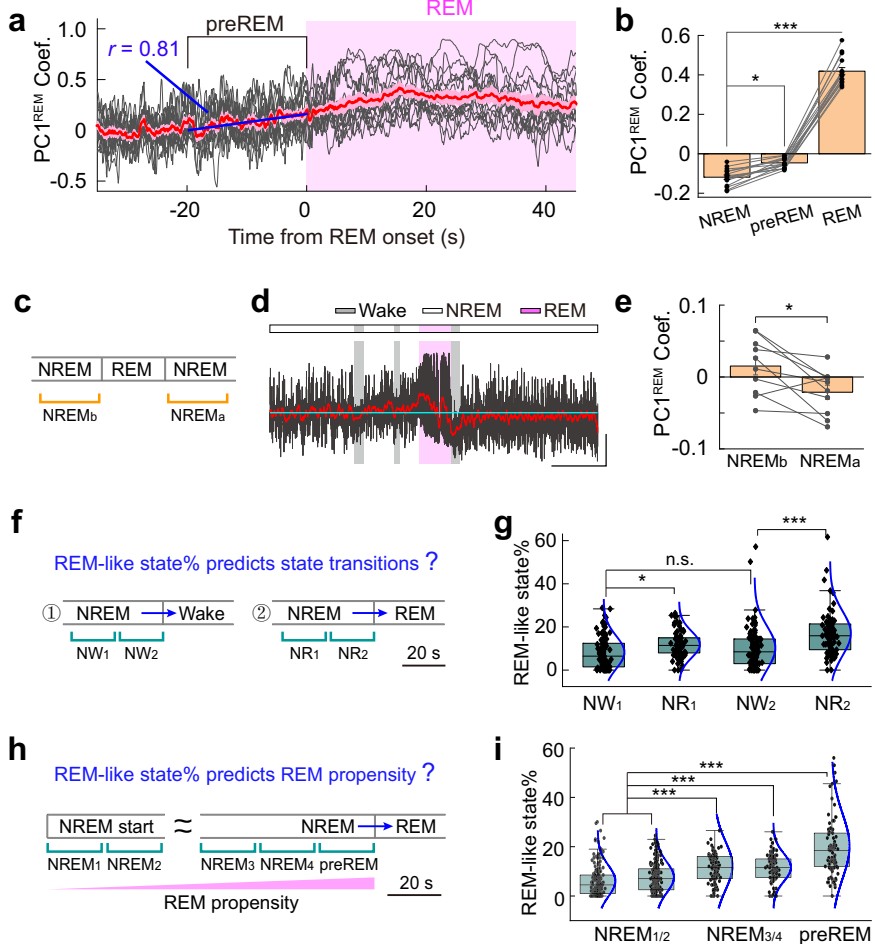

**Fig. 6 | Activity in the occipital cortex signifies REM sleep. a** Dynamic change of the REM-like state during NREM-to-REM transitions. Gray line, data from individual recordings. Red line, an average of 15 recordings from 5 mice. Shading is SEM. Blue line, a linear fit of the average data during preREM (the 20-s immediately before REM onset). **b** Quantification of PC1$^{REM}$ dynamics during NREM-to-REM transitions. $P = 0.036$ for NREM vs. preREM; $P < 0.0001$ for NREM vs. REM; Two-tailed paired $t$-test. $n = 15$ recordings from 5 mice. Data are mean ± SEM. **c** Diagram showing different NREM periods. **d** An example showing dynamic changes of PC1$^{REM}$ before and after a REM bout. Scale, 0.5 and 100 s. **e** Quantification of PC1$^{REM}$ before and after REM bouts. $P = 0.011$; Two-tailed paired $t$-test; $n = 10$ recording from 5 mice. **f** Diagram describing the definition of different periods of NREM before the NREM → Wake or the NREM → REM transitions. **g** Box-plot showing the quantification of REM-like brain state in different NREM periods. Box, 25–75% range and median line; Whisker, 1.5 interquartile range; Blue line, data fitting with a normal distribution model. $F = 14.5$; $P = 0.41$ for NW$_1$ vs. NW$_2$; $P = 0.043$ for NW$_1$ vs. NR$_1$; $P < 0.001$ for NW$_2$ vs. NR$_2$. one-way ANOVA with posthoc Tukey's tests; $n = 85$ and 74 (15 recordings from 5 mice) for NREM → Wake and NREM → REM transitions, respectively. **h** Diagram describing the definition of different periods of NREM that are associated with the different propensities of transitioning into REM sleep. **i** Box-plot showing the quantification of REM-like brain state in different NREM periods. The box-plot elements were same as **g**. $F = 48$; $P = 0.48$ and 0.98 for NREM$_1$ vs. NREM$_2$ and NREM$_3$ vs. NREM$_4$, respectively; $P < 0.0001$ for all other paired-wised comparisons; one-way ANOVA with posthoc Tukey's tests; $n = 81$ REM bouts and 166 NREM bouts, 15 recordings from 5 mice. Raw data for **b**, **e**, **g**, **i** and exact $P$ values are provided in a Source Data file.

The infra-slow sigma oscillation is thought to represent a periodic increase of arousal level during NREM[42,43], which may lead to micro-arousal events. We thus next analyze the relation between the REM-like activity and microarousal events. We found that there were more REM-like states before microarousal events and more REM-opponent events during microarousal events ($P < 0.001$ for both comparisons, one-way repeated measures ANOVA with posthoc Tukey's test; Supplementary Fig. 17).

**Occipital Activity Signifies NREM to REM Transition**

We have shown that cortical activity during REM sleep was dominated by occipital activation, and the NREM activity pattern oscillated between REM-like and REM-opponent states, raising a possibility that the occipital activation may signify the occurrence of REM sleep. To test this idea, we analyzed global cortical activity during NREM-to-REM transitions. We found that there were significantly more REM-like states immediately before the NREM-to-REM transition (preREM,

defined as the 20-sec before the transition; $P < 0.05$; Paired $t$-test; Fig. 6a, b), and the occurrence of REM-like pattern increased linearly during preREM (Pearson's $r = 0.81$, $P < 0.0001$; Fig. 6a). Additionally, the probability of REM-like states was significantly lower during the NREM period following prolonged REM bouts ($P = 0.011$; Paired $t$-test, $n = 11$ recordings from 7 mice) (Fig. 6c–e), suggesting that the REM-like activity pattern during NREM sleep may represent the propensity of brain state transition into REM sleep. Indeed, the occurrence of the REM-like state was associated with whether the brain would transit into wakefulness or REM sleep. There were significantly more REM-like states before the NREM-to-REM transitions ($P < 0.05$, one-way ANOVA with Tukey's posthoc tests; Fig. 6f, g).

The association of REM-like activity patterns with the propensity of REM sleep was further supported by our additional analysis, in which we measured the probability of the REM-like state during different NREM periods that were associated with the different propensity of transitioning into REM sleep. We found that the cortical

pattern showed significantly more REM-like states during the later stages of NREM bouts when there was a high propensity of transitioning into REM sleep, compared with the early stages ($P < 0.0001$, one-way ANOVA with Tukey's posthoc tests; Fig. 6h, i). Together, these results indicated that the occurrence of the REM-like activity pattern could signify the propensity of brain state transition from NREM to REM sleep.

## Modulation of NREM to REM switching by occipital activity

We have shown that the occipital cortex was highly active during REM sleep, and such an activity pattern was also associated with the propensity of REM sleep. We thus hypothesized that neural activity in the occipital cortex might play a role in regulating REM sleep. To test this idea, we suppressed activity in the occipital cortex via optogenetic exciting the GABAergic neurons in these regions using GAD2-Cre mice[44] with AAV-mediated expression of red-shifted channelrhodopsin, ChrimsonR[45] (Fig. 7a). To achieve a large area of optogenetic manipulation and minimize damage to the cortex by inserting optical fibers (Supplementary Fig. 18a, b), we applied light through the transparent skull using a head-mounted high-power LED (567 nm; 10 mW, 20 Hz, 2 min/trial, randomly applied every 5–7 min; Fig. 7a and Supplementary Fig. 18a, c). We found that light application significantly reduced the probability of REM sleep (mean, −59.6% of baseline; 95% Cis, [−45.8%, −73.4%]; $P < 0.001$, bootstrapping), increased NREM sleep (mean, 12.6% of baseline; 95% Cis, [8.4%, 16.8%]; $P < 0.001$, bootstrapping), and had no apparent effect on wakefulness ($P = 0.38$, bootstrapping; Fig. 7b). ($n = 25$ testing sessions from 5 mice expressing ChrimsonR, 5 sessions/mouse). The stimulation-induced REM suppression was observed in all five mice tested in our experiment (Supplementary Fig. 19), indicating the robustness of this effect. This result was unlikely to be a non-specific effect of the light since we detected no light-induced changes in all three brain states in mice expressing a control virus (Fig. 7b, c and Supplementary Fig. 19; $P > 0.62$ for all three states, bootstrapping; $n = 24$ testing sessions from 5 mice expressing mCherry, 4 or 5 sessions/mouse). The laser stimulation-induced modulation in the ChrimsonR- and mCherry-expressing mice were also compared using two-way repeated measures ANOVA, which revealed significant main effects of laser stimulation on the percentage of both REM and NREM sleep but not wakefulness (Wake, $F_{(1, 8)} = 0.097$, $P = 0.76$; NREM, $F_{(1, 8)} = 13.0$, $P = 0.007$; REM, $F_{(1, 8)} = 25.5$, $P < 0.001$; two-way repeated measures ANOVA).

The reduction in REM sleep was not because we could not identify REM sleep during the stimulation period—EEG signals for both REM and NREM sleep during the stimulation period showed characteristic θ and δ oscillations, respectively (Supplementary Fig. 20). Furthermore, optogenetic stimulation did not produce unnatural brain states, as the EEG spectrum (0.5–18 Hz) for both REM and NREM sleep during the stimulation period had no significant difference from that during the no-stimulation period (REM, $P = 0.96$; NREM, $P = 0.68$; Kolmogorov–Smirnov test; Fig. 7d).

In the optogenetic silencing experiments, we observed stimulus-induced reductions in REM sleep, which may be the result of reduced REM entry or reduced REM maintenance. Brain state transition analysis can distinguish these possibilities. For a given time point (5-s bin), brain state transition analysis determines the probability of the current brain state transitioning to other brain states. In this analysis, we compared the transition probability during the baseline and stimulation periods (Fig. 7e, f and Supplementary Fig. 21). Each bar in the figure represents the transition probability averaged in one minute. We found that the primary effect of the light stimulation was to decrease the transition from NREM to REM while stabilizing NREM sleep (increased NREM→NREM transition) (NREM→REM, $P < 0.001$; NREM→NREM $P < 0.001$; Paired $t$-test or Wilcoxon signed-rank test, see Methods section; Fig. 7e). This result was consistent with our observation that the REM bout number was markedly decreased during the stimulation period (ChrimsonR: baseline, $26.2 \pm 1.8$;

stimulation: $8.8 \pm 1.4$; $P < 0.0001$, Paired $t$-test; mCherry, baseline, $23.8 \pm 3.1$; stimulation: $18.4 \pm 1.3$; mean ± SEM; $P = 0.16$, Paired $t$-test; Supplementary Fig. 22). There was also a tendency for decreased REM consolidation and increased transition from REM to wakefulness (REM → REM, $P = 0.069$; REM → Wakefulness, $P = 0.062$; Paired $t$-test; Fig. 7e). Together, these results indicate that neural activity in the occipital cortex plays a role in controlling the NREM-to-REM transition.

We next tested whether neural activity in the occipital cortex can bidirectionally modulate the transition between NREM and REM. Since excessive activation of cortical excitatory neurons can cause seizure, we only expressed ChrimsonR (driven by CaMKII promotor) in a small region of the occipital cortex, the RSP (Fig. 7g). Light stimulation (638 nm, 0.3–2 mW at fiber tip, 20 Hz, 2 min/trial, randomly applied every 5–7 min) caused a significant increase of REM sleep ($P = 0.0028$, Paired $t$-test) and decrease of NREM sleep ($P = 0.007$, Paired $t$-test), and no detectable change in wakefulness was observed ($P = 0.33$, Paired $t$-test) (Fig. 7h). The main effect of the light stimulation was to increase the transition from NREM to REM and decrease the consolidation of NREM sleep ($P < 0.001$ for both transitions; Paired $t$-test; Fig. 7i), reverse mirroring the effect of inhibiting the occipital cortex.

Taken together, we have demonstrated that occipital activity signifies REM propensity and plays an active role in controlling REM sleep. An immediate intriguing question is whether inhibition of occipital activity leads to an increase in REM pressure since REM sleep is also homeostatically regulated[46,47]. In our optogenetic inhibition experiment, we did observe REM rebound shortly after the laser stimulation (Fig. 7b and Supplementary Fig. 19), suggesting the possibility of increased REM pressure by suppressing occipital activity. To further examine this, we performed a new experiment in which we inhibited occipital activity for a longer time (Laser on, 20 min; Laser off, 8–10 min) and with higher laser power (20 mW, to achieve more significant inhibition; Fig. 7j). We found that prolonged inhibition of occipital activity effectively suppressed REM sleep throughout the stimulation period—the percentage of REM sleep decreased to 3.5% compared with 10.5% before stimulation ($P = 0.032$, one-way ANOVA with posthoc Tukey's test; Fig. 7j–l). Importantly, mice frequently entered into REM shortly after stimulation, resulting a substantial increase in the percentage of REM sleep (25.9%; $P < 0.001$, one-way ANOVA with posthoc Tukey's test, compared with baseline; Fig. 7j–l). These results show that inhibition of occipital activity produces REM sleep pressure, suggesting that the occipital cortex plays a role in REM homeostasis.

## Discussion

Using mesoscale Ca²⁺ imaging from the entire dorsal cerebral cortex in mice, we revealed highly structured cortical activity and distinct global activation patterns during the sleep-wake cycle, with more activation in the somatic sensorimotor network during wakefulness, and more active medial cortical network, especially the RSP, during REM sleep (Figs. 1e and 2–4). In particular, we uncovered a shared global cortical activity pattern for REM and NREM sleep control, as is illustrated by the diagram shown in Fig. 8. REM sleep is dominated by the cortical activity pattern with higher activation in the occipital cortex and low activity in the somatic sensorimotor cortex (the REM-like activity pattern), and such dominance gradually emerges during the NREM-to-REM transition. The cortical activity during NREM sleep oscillates between the REM-like pattern and the REM-opponent pattern. Moreover, the occurrence of the REM-like pattern signifies the propensity of brain state transition from NREM to REM sleep. Inhibition of the REM-like pattern by suppressing neural activity in the occipital cortex can retain the brain state in the REM-opponent state, thus preventing the transition from NREM to REM sleep and promoting NREM sleep (Fig. 7).

During sleep, although our body is relatively inactive, our brain is very active. Our direct imaging of global neural activity in the entire

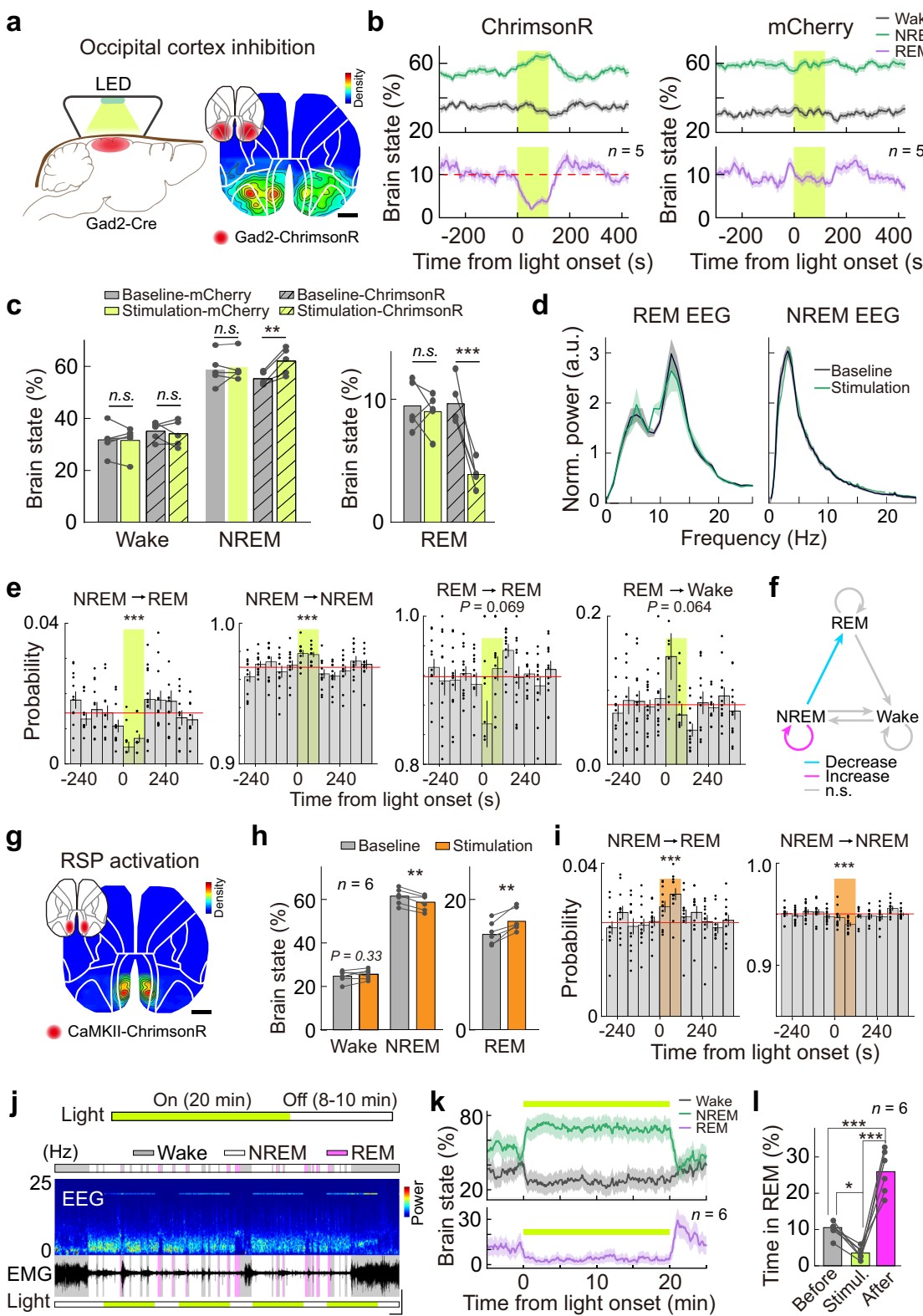

dorsal cortex with a high spatiotemporal resolution shed light on understanding the global organization of cortical activity patterns during sleep. Brain activity during REM sleep was thought to resemble that during wakefulness, at least at the EEG level[48,49]. However, our measurement of global cortical activity suggests a fundamental difference between the two states, with a large difference in the activation patterns of the somatic sensorimotor network and the medial cortical network, and the cortical activity patterns during NREM sleep were

more similar to the REM activity than that between wakefulness and REM.

Our results also showed that the cortical network during NREM sleep oscillates between the REM-like state and the REM-opponent state on a rapid time scale, demonstrating the heterogeneous cortical activity patterns during NREM sleep. Conceptually, our finding is consistent with the concept of "local sleep"[50], which refers to the locally generated neuronal ceasefire during wakefulness, especially

**Fig. 7 | Control of REM sleep by the occipital cortex. a** Schematic of experiment and region of virus expression. **b** Percentage of each brain state during light stimulation (yellow shading). $n = 25$ sessions from 5 mice (5 sessions/mouse) for ChrimsonR group and 24 sessions from 5 mice (4 or 5 sessions/mouse) for mCherry group. Shading, SEM **c** Percentage of each brain state during stimulation and baseline. Wake, $F(1, 8) = 0.097$, $P = 0.76$; NREM, $F(1, 8) = 13.0$, $P = 0.007$; REM, $F(1, 8) = 25.5$, $P < 0.001$; ChrimsonR group: $P = 0.74$, 0.0017, and 0.00017 for Wake, NREM, and REM, respectively; mCherry group: $P = 0.46$, 0.65, and 0.60 for Wake, NREM, and REM, respectively; Two-way repeated measures ANOVA with Tukey's posthoc test. **d** Normalized EEG power spectrum during light stimulation and baseline. Shading, SEM. $P = 0.98$ and 0.68 for REM and NREM, respectively; Kolmogorov-Smirnov test. $n = 22$ or 25 sessions from 5 mice for REM and NREM, respectively. **e** Transition probability for indicated brain states during light stimulation (shading). Red line, baseline transition probability. $n = 25$ sessions from 5 mice. $P < 0.001$ for both NREM→REM and NREM→NREM transitions; Paired $t$-test or Wilcoxon signed-rank test. $P = 0.069$ and 0.064 for REM→REM and REM→Wake transitions, respectively; Paired $t$-test or Wilcoxon signed-rank test. Data are mean ± SEM. **f** Diagram summarizing brain state transitions during light stimulation. **g** Schematic of experiment and region of virus expression. **h** Percentage of each brain state during stimulation and baseline. $P = 0.33$, 0.007, and 0.0028 for Wake, NREM, and REM, respectively; Paired $t$-test. $n = 42$ sessions from 6 mice. **i** Transition probability of NREM→REM and NREM→NREM during light stimulation (shading). $n = 42$ sessions from 6 mice. $P < 0.001$ for both transitions; Paired $t$-test. Data are mean ± SEM. **j** Schematic of experiment showing prolonged optogenetic suppression of the occipital cortex and an example recording (scale, 0.5 mv and 5 min). **k** Percentage of each brain state during light stimulation. $n = 30$ sessions from 6 mice (5 sessions/mouse). Shading, SEM **l** Percentage of REM sleep before, during, and after light stimulation. Before vs. Stimuli., $P = 0.032$; After vs. Stimuli., $P < 0.0001$; Before vs. After., $P < 0.0002$; One-way repeated measures ANOVA with Tukey's posthoc test. $n = 30$ sessions from 6 mice. Raw data for **c–e**, **h–j**, and **l** are provided in a Source Data file.

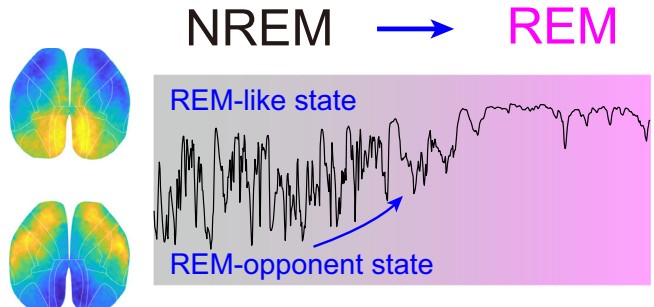

**Fig. 8 | Diagram illustrating cortical activity during NREM-to-REM transition.** The cortical activity pattern during REM sleep is dominated by elevated activation in the occipital cortex and low activity in the somatic sensorimotor cortex (the REM-like pattern), and such dominance gradually emerges during the NREM-to-REM transition. The cortical activity during NREM sleep oscillates between the REM-like pattern and a REM-opponent pattern. The occurrence of the REM-like pattern is associated with the propensity of brain state transition from NREM to REM sleep. Inhibiting the REM-like pattern by suppressing neuronal activity in the occipital cortex can retain the brain state in the REM-opponent state, thus preventing the transition from NREM to REM sleep.

when sleep pressure is high. While local sleep represents spatially localized micro-sleep and its occurrence is associated with increased sleep pressure, our finding indicates that transient REM-like states embed in NREM sleep, and the occurrence probability of such state is associated with transition probability from NREM to REM, resembling the "REM pressure" of the brain[51]. Therefore, both studies emphasized that a specific brain state can have characteristic features of other brain states in the space or the time domain.

Neural activity during sleep has been proposed to associate with various brain functions, including memory consolidation[52–57]. Our mesoscale optical imaging of cortical activation provides new insights into how these activities are spatiotemporally organized in the dorsal cortex. The PGO wave-like activation of the RSP during REM sleep implicates its association with REM functions. In fact, RSP has been shown to be closely related to the memory process[58]. Our results offer a potential explanation of the RSP involvement in memory, which is also consistent with the memory role of the PGO waves[39,40] and REM sleep[59,60]. The high activation of the occipital cortex during REM sleep is also consistent with a human EEG study, which shows that the 'dreaming' experience is tightly correlated with activation of the occipital "hot zone"[61].

The RSP is part of the brain default mode network (DMN), which is more active when a subject is not interacting with the outside world[62]. With this in mind, it is not difficult to understand that RSP is relatively inactive during the waking period, because the main purpose during wakefulness is to interact with the outside world. While interestingly, the RSP is also less active during NREM sleep but highly active during REM sleep, indicating that the highly active brain during REM sleep is more internally orientated.

The neural control of sleep-wake state switching has been thought to be a privilege of the subcortical network, especially the brainstem and hypothalamus[6–8,63]. The main role of the cortex in sleep-wake regulation is thought to generate homeostatic sleep pressure in an activity-dependent manner[10,50,64], or to support arousal[9] or promote sleep[11] through top-down feedback connections to subcortical regions. For REM sleep control, it is generally accepted that the brainstem plays a central role, with significant modulation by the hypothalamus[6–8,38,65–68]. Our finding that activity in the medial cortical network is also required for REM sleep uncovers an unexpected role of the cortex in REM control. These results provide evidence supporting the control of the sleep-wake cycle by the fast cortical activity dynamics, and indicate that REM sleep, as a unique state of the entire brain network, requires coordinating neural activation within the whole-brain network. Conceptually, our results showed that the structured cortical activity during sleep is not just an epiphenomenon of various sleep-wake states, but also plays an active role in regulating sleep state switching.

## Methods

### Experimental model and subject details
All experimental procedures followed the National Institutes of Health guidelines and were approved by the Animal Care and Use Committee at the Institute of Neuroscience, Chinese Academy of Sciences. Both male and female mice (>8 weeks at the time of surgery) were used. Mice were housed in rooms (temperature: $23 \pm 1°C$; humidity: 50–70%) under a 12/12-hr light/dark cycle (light on at 7 a.m.) with ad libitum access to food and water. Mice with implants for widefield imaging or optogenetic manipulation were housed individually. Wild-type mice (C57BL/6) (six male) were purchased from institute-approved vendors (Shanghai Silaike or LingChang Experiment Animal Co., China); *GAD2-IRES-Cre* (six male and ten female) and *Thy1-GCaMP6s* (eight male) mice (stock #: 010802 and 024275, respectively) were obtained from Jackson Laboratory.

### Surgical procedures
For widefield imaging experiments, the *Thy1-GCaMP6s* mice were anesthetized with isoflurane (5% for induction; 1.5–2% for maintenance) and placed on a stereotaxic frame with a heating pad. A "transparent skull" was prepared for chronic optical imaging using a similar procedure described previously[21]. The scalp was removed, and the skull was cleaned using hydrogen peroxide solution. The entire dorsal part of the skull (except that above the olfactory bulb and the cerebellum) was thinned using a dental drill and covered with a thin

layer of cyanoacrylate glue (ZAP-A-GAP). Two stainless steel screws (M1.0 × 3 mm) were inserted into the skull above the auditory cortex (AP −3.0 mm, ML 4.6 mm) of both hemispheres, and another two screws were inserted into the skull above the olfactory bulb (AP 4.9 mm, ML 0.8 mm) and the cerebellum (AP −6.0 mm, ML 1.0 mm), respectively. A thin wire (stainless steel wire (0.1 mm in diameter) insulated with polyimide tubing (0.2 in diameter)) was attached to one screw above the auditory cortex to record EEG. Another two wires were inserted into the neck muscle to record EMG. The ground wire was attached to the screw above the cerebellum. Finally, a customized head-plate was cemented to contact the cerebellum. A 3D-printed cap was cemented to protect the transparent skull, and the cap also served as a light-shielding cap during imaging. The dental cement used to secure the implant was mixed with carbon to minimize light leakage during imaging.

For pontine LFP recording, electrodes (FeNiCr wires, 50 μm in diameter, California Fine Wire) were implanted in the pons (AP, −5.4 mm, ML, 0.5 mm, DV 4.5 mm from the brain surface).

For optogenetic inhibition experiments, mice were prepared by using a similar surgical procedure as described above. To express the AAV virus, we made small craniotomies (−0.5 mm in diameter) on top of the retrosplenial cortex (AP −3.4 mm, ML 0.7 mm, DV 0.5 mm from the cortical surface) and the visual cortex (injection site 1: AP −3.1 mm, ML 2.5 mm, DV 0.4 mm from the cortical surface; injection site 2: AP −4.0 mm, ML 2.2 mm, DV 0.4 mm from the cortical surface), and injected the virus (0.3 μl/injection site) using Nanoject II (Drummond Scientific) via a glass pipette (23 nl/injection; inter-injection interval, 15−30 s).

For the RSP activation experiments, AAV-CaMKII-Cre and AAV-FLEX-ChrimsonR were 1:1 mixed and injected (0.3 μl/injection site) using the following coordinate: AP −3.3 mm, ML 0.5 mm, DV 0.5 mm from the cortical surface.

The following AAV viruses were used in current study: In optogenetic inhibition experiments, AAV2/9-hSyn-FLEX-ChrimsonR-tdTomato (Titer: $3.33 \times 10^{12}$ v.g./ml; Shanghai Taitool Bioscience Co., China); AAV2/9-EF1α-DIO-mCherry-WPRE-pA (Titer: $2.6 \times 10^{12}$ v.g./ml; BrainVTA., China). In optogenetic activation experiments, AAV2/9-hSyn-FLEX-ChrimsonR-tdTomato (Titer: $5.0 \times 10^{12}$ v.g./ml; Shanghai Taitool Bioscience Co., China); AAV2/9-CaMKII-Cre (Titer: $2.23 \times 10^{12}$ v.g./ml; BrainVTA., China).

## Widefield imaging

Imaging experiments were performed on head-fixed mice. Mice were habituated to the head-fixed apparatus starting one week after surgery. The head-fixed device consists of a head plate holder, a low-profile lightweight disc (made of carbon fiber), and an optical breadboard covered with bearing balls. During experiments, mice were placed into the disc and attached to the head plate holder, and they could easily move the disc around, thus exhibiting less stress. Each mouse was first handled for three days and then restrained to the head-fixed device daily. The duration of the restrain started from 10 min and gradually increased to 3−4 h until they could reach stable sleep. A camera was used to capture a facial video of the mouse during the habituation, and the procedure was stopped if there were signs of excessive stress.

During the imaging experiments, an infrared camera (Hikvision) with the illumination of 840 nm or 920 nm was used to capture a movie of the facial movements and pupils of the mice. The frame rate of the video was 25 Hz. The camera could also capture part of the excitation light from the macroscope, and we used this information to synchronize the behavior video with the Ca²⁺ imaging data.

Imaging was performed with a custom-built fluorescence macroscope using parts from Olympus and Thorlabs. The macroscope consisted of a zoom body of the Olympus MVX10 microscope with a 2x objective lens (MVPLAPO 2XC), a GFP filter

set (Thorlabs), excitation LEDs (470 nm and 405 nm, Thorlabs), and a CCD camera (Qimaging, Retiga R1). All other optomechanical components used to build the macroscope were purchased from Thorlabs. For imaging experiments with both 470 nm and 405 nm excitations, alternating 470 nm and 405 nm illumination was controlled using an Arduino board triggered by the exposure-out of the camera. Images were acquired using Micro-manager software (NIH, ver: 1.4) with a frame rate of 10 Hz.

The light level in the recording box was ~80 lux, and the light level at the mouse eye was <10 lux (light is blocked by the imaging lens).

For imaging experiments with pontine LFP recording, LFP (high-pass filtered at 0.5 Hz and digitized at 1 kHz) was acquired using an RHD acquisition board from Intan Technologies.

## Polysomnography recordings

For polysomnography recordings during widefield imaging, the EEG and EMG signals (high-pass filtered at 0.5 Hz and digitized at 1 kHz) were acquired using an RHD acquisition board and RHX software (Ver: 3.03) from Intan Technologies. The acquisition board was also used to record the exposure-out signal from the camera to synchronize the EEG/EMG signals with the imaging data.

For polysomnography recordings in optogenetic manipulation experiments, mice were transferred into recording cages, placed in a sound-attenuation box (80 × 100 × 120 cm), and connected to the amplifier (TDT system-3 amplifier RZ2 + PZ5) with a flexible recording cable via a commutator. EEG/EMG was high-pass filtered at 0.5 Hz and digitized at 1526 Hz. Mice were habituated for two days before recording. During recording, a large power LED (for optogenetic silencing experiments) or optical fibers (for optogenetic activation experiments) were attached to the head implant. The recording session started after 30 min and lasted for three hours.

## Optogenetic inhibition

To achieve a large area of light illumination, we attached a high-power LED (567 nm, with a SinkPAD-II 20 mm base) (QUADICA) on the top of the 3D printed cap, and covered it with light-proof copper foil. The LED was powered using a T-Cube LED driver from Thorlabs, which was controlled by the TTL output of an RZ5 system (Tucker-Davis Technologies). The RZ5 system and OpenEx (Tucker-Davis Technologies, Ver: 2.31) were also used to record the EEG/EMG signals during the optogenetic manipulation experiments.

All experiments were performed during the daytime between 10 a.m. and 6 p.m., and each test session lasted for 3 h. For the 2-minute occipital cortex inhibition experiments, in each session, we applied 2-min of light stimulation (pulse duration: 5 ms; pulse interval: 45 ms; 21 trials/session) with a time interval of 5−7 min. For the occipital cortex inhibition experiments, the light power measured at the surface of the skull is 10 mW, which is equivalent to 0.004 mW when delivering light through an optical fiber with a diameter of 200 μm. Each mouse was tested for five sessions (1 session/day). For the 20-min occipital cortex inhibition experiments, we applied 20-min of light stimulation (pulse duration: 5 ms; pulse interval: 45 ms; 5 trials/session) with a time interval of 8−10 min in each session. The light power measured at the surface of the skull is 20 mW, equivalent to 0.008 mW when delivering light through an optical fiber with a diameter of 200 μm. Each mouse was tested for five sessions (1 session/day).

For the RSP activation experiments, optical fiber (diameter: 200 μm; N.A.: 0.37) was placed bilaterally on the top of the thinned skull above virus injection sites. The light power was measured at the tip of the optical fiber. Each mouse was tested for 5 or 8 sessions (1 session/day). In each session, we applied 2-minutes of light stimulation (pulse duration: 5 ms; pulse interval: 45 ms; 20 trials/session) with a time interval of 5 − 7 min. The light power measured at the surface of the skull is 0.3−2 mW.

## Histology and immunohistochemistry

To verify the virus expression, mice were deeply anesthetized and immediately perfused using 0.1 M PBS, followed by 4% PFA. The brain tissues were removed and post-fixed overnight in 4% PFA before dehydration in a 30% sucrose solution. Brain samples were embedded with OCT compound (NEG-50, Thermo Scientific) and cut into 50-μm sections using a cryostat (HM525 NX, Thermo Scientific). Brain sections were washed in PBS and coverslipped with mounting media.

The fluorescence images were captured using an epifluorescence microscope (VS120, Olympus).

## Analysis of widefield imaging data

**Preprocessing.** The imaging was performed using a camera binning of $4 \times 4$ or $2 \times 2$, and saved to a Tiff stack. The image stack was downsampled by a factor of 2 if $2 \times 2$ binning was used, thus, the final image used in all following analyses had a resolution close to $260 \times 256$. The image stack was converted to an unformatted binary file to facilitate file I/O in later processing. Previous reports showed that a registration step is required to correct motion-induced movements of the images[69,70]. However, in our data, we found that the motion-induced movements were negligible as no obvious blurring was found in the time-averaged images (typically from 40,000 images recorded in about 1 h; see an example in Fig. 1b).

The imaging data were transformed into the change in fluorescence ($\Delta F/F_0$) using a baseline computed by averaging all images in a recording (40,000 images). In a small fraction of our experiments, we found a slight decrease in the overall fluorescence intensity which was likely caused by bleaching (especially when the control excitation of 405 nm was used). In these cases, we computed the mean fluorescence intensity of each frame and fitted the change in the mean intensity using a second-order exponential function. The fitting result was then used to correct the slow signal decay caused by bleaching.

The global signal in the imaging data, which is signal fluctuations common to the whole brain[69,71], has been reported to associate with non-neuronal physiological artifacts, such as respiration and hemodynamics[72]. We thus removed the global signal from the $\Delta F/F_0$ data via linear regression[69,71], and this process is termed global signal removal (GSR). The global signal in our imaging data had a high similarity with the first PCs when PCA was performed before GSR was implemented, and in such conditions, PC1 showed a global activation, confirming that GSR removed global correlation in the imaging data.

In order to minimize the influence of blood vessels (especially large veins), we generated a mask to remove large blood vessels in the imaging data. To generate the mask, we first enhanced blood vessels in the time-averaged image using a filter (https://github.com/timjerman/JermanEnhancementFilter/blob/master/vesselness2D.m), then created a mask using 'imbinarize', a build-in function in MATLAB, using the 'adaptive' method with 'Sensitivity' = 0.5. Finally, the mask was superimposed with the standard deviation of the image stack, and manual ROIs were used to remove artifacts caused by remaining blood vessels. The mask and manual ROIs were merged to get the final mask.

The blood vessel mask was applied to the $\Delta F/F_0$ data before further processing was performed.

**Regional map generation.** In order to match the imaging data with an anatomical atlas, we registered the imaging data to a reference atlas (the Allen Mouse Brain Atlas)[22,73]. First, a flattened cortical map was generated from the reference atlas, and the map was manually trimmed to fit our imaging data by removing a fraction of the most posterior and lateral regions in the neocortex. Such an operation is commonly used in previous studies[22,69,74]. We then merged small anatomy regions in the map and got the final reference regional map with nine brain regions, including the primary motor cortex (MOs), the secondary motor cortex (Mop), the somatosensory cortex, barrel field (SSb), the somatosensory cortex, upper limb (SSu), the somatosensory

cortex, lower limb (SSl), the primary visual cortex (VISp), the association visual cortex (VISs), the dorsal retrosplenial cortex (RSPd), and the lateral retrosplenial cortex (RSPl). Only a small part of the auditory cortex can be imaged using the current method; we thus did not include the auditory cortex.

For each recording, we registered the reference map onto the imaging data using the registration package, ANTs (http://stnava.github.io/ANTs/). The registration was performed in three steps: 1) The time-averaged image was binarized using a combination of adaptive thresholding and manual ROIs; 2) The reference map was manually scaled to match the imaging data grossly. This step was performed to facilitate the following registration using ANTs. 3) Automatic, elastic registration was performed in ANTs using a similarity metric of neighborhood cross-correlation.

**Brain atlas-based quantification.** To quantify the $\Delta F/F_0$ for each brain region, we first averaged the $\Delta F/F_0$ for each brain state and then computed the mean $\Delta F/F_0$ in each brain region using a region mask after the removal of blood vessels (Supplementary Fig. 4). The results for each recording were then z-score normalized across the nine brain regions and generated plotting data for Fig. 1e.

A similar quantitation procedure was used to quantify the distribution of the PCs or activity pattern and generated plotting data for Supplementary Figs. 5b, 8c,10b, and 16.

**Correlation matrix and hierarchical clustering.** We first divided the brain into a $16 \times 16$ grid (the size of each patch is $\sim 10 \times 10$ pixels) and computed all pairwise correlations using the 'corrcoef' function in MATLAB to obtain a correlation matrix of neural activity in each brain state. We then used the 'clustergram' function in MATLAB with default parameters to perform hierarchical clustering, and grouped brain regions with similar activity dynamics. The regional cluster was generated using a distance of 5. Clusters containing fewer than four patches were merged into the nearest cluster.

To generate the anterolateral and occipital network, we divided cortical regions into two groups using hierarchical clustering and averaged the results from 15 recordings to get the map in Fig. 2c.

**PCA analysis.** PCA was applied to the blood vessels removed $\Delta F/F_0$ data to decomposite the imaging data and extract activity patterns using the built-in function in MATLAB. For PCA across the entire recording, the imaging data were temporally down-sampled with a decimating factor of 2 or 5 to reduce the computational load. No down-sampling was used when PCA was used to analyze data during REM. The resulting PCs were then inpainted to fill regions removed by the blood vessel mask, using a built-in function 'inpaintExemplar' in MATLAB with 'PatchSize' = 5. The inpainted PCs were then spatially smoothed using an averaging filter of $5 \times 5$. Finally, a mask derived from the region map was applied to the PCs to remove pixels outside of the brain.

Because the PCs can be of either one of the polarities, we reversed the polarity of the major PCs (as well as the corresponding coefficient) in some cases, such that a positive coefficient always corresponded to an increase in the fluorescent signals.

**Analysis of PC events.** We first computed an adaptive baseline of the coefficient (sampling rate, 5 Hz) of each PC using a moving average (20 s), detected signals above 1x standard deviation from the baseline, and used the peak of these signals to determine each event (from 2 s before to 2 s after the peaks). We then used the Gaussian function to fit each event and used the sigma of the Gaussian curve as the half-width of each event.

The distribution of the event width for each recording was used to generate Fig. 3e, and this distribution was then fitted using the Gaussian function to obtain a mean event duration for each recording.

**Analysis of PC1$^{REM}$-like pattern during NREM and Wakefulness.** The cross-correlation (CC) between each image frame and PC1$^{REM}$ was calculated using the 'corrcoef' function in MATLAB. The CC in each brain state was averaged to get data for Fig. 5b. The REM-like events and REM-opponent events were defined as image frames with a CC value >0.4 or <−0.4, respectively. The neocortical patterns in Fig. 5c were an average of 5 events as marked on the CC plot. The occurrence probability of the two events during the defined NREM periods was used to generate Fig. 5d.

The CC was also used to analyze the relation between PC1$^{REM}$-like pattern and the infra-slow sigma oscillations or microarousals. EEG sigma oscillation (9–17 Hz) was computed according to previous work[42] for manually selected NREM bouts with no microarousals. We calculated the EEG sigma power (9–17 Hz) every 1 s and then computed the correlation with smoothed CC (using a 10-s window). Microarousal events were identified manually as periods with a brief increase of EMG power during NREM sleep.

**Correlation between PC1$^{REM}$ and EEG θ/δ.** We first computed a spectrogram for EEG using a fast Fourier transform (FFT) with a frequency resolution of 0.18 Hz, and calculated the ratio between the θ band (6–10 Hz) and δ band (0.5–4 Hz). We then averaged the ratio and the coefficients of PC1$^{REM}$ in each 5-s window and computed the correlation (Fig. 4e).

**Correlation between PC1$^{REM}$ and eye movements.** To extract the eye movements, we used the FaceMap software (www.github.com/MouseLand/FaceMap)[75]. There was a high correlation between eye movements in the horizontal direction and vertical direction; we thus only used the horizontal movements for all analyses. To detect bouts of eye movements, we first computed an adaptive baseline using a moving average of 500 points, and detected signals that were 1x standard deviation higher than the baseline. We then used the onset of each eye movement to align the coefficients of PC1$^{REM}$ in each recording. The mean coefficient was z-score normalized and averaged between different animals (Fig. 4f).

To analyze the correlation between occipital activity and eye movements during wakefulness, we manually identified each eye movement event.

**Detection of P-waves.** P-waves were detected as large negative peaks according to previous work[41]. We used the MATLAB function 'findpeaks' to extract events greater than 4x standard deviation of the pontine LFP during REM sleep. The detected events were used to align the coefficients of PC1$^{REM}$ in each recording. The z-score normalized coefficients were averaged across animals.

**Detection of tonic REM vs. phasic REM.** Tonic and phasic REM were defined based on the presence of muscle twitches according to previous work[37,76–78]. We used the FaceMap software (www.github.com/MouseLand/FaceMap) to extract facial movements and calculate the mean of the signal during REM. Phasic REM was defined as REM with greater than average facial movements, and tonic REM was defined as REM with less than average facial movements.

**Brain states scoring.** To score the brain states using EEG/EMG signals, we performed spectral analysis on the EEG using a fast Fourier transform (FFT) with a frequency resolution of 0.18 Hz. The brain states were scored every 5 s semi-automatically using a MATLAB GUI and validated manually by trained experimenters. Brain states classification was performed according to established criteria:[25,79] Wakefulness was defined as desynchronized EEG and high EMG activity; NREM sleep was defined as synchronized EEG with high-amplitude δ activity (0.5–4 Hz) and low EMG activity; REM sleep was defined as high power at θ frequencies (6–10 Hz) and low EMG activity.

We also divided wakefulness into active and quiet wakefulness based on whether the facial movements of mice were greater or less than average movements. Please note that the current definition is different from the classical definition that uses body movements. We used facial movements because we only recorded a video of the face of the head-fixed mice.

**NREM/REM EEG spectrum during light stimulation.** We first computed a spectrogram for EEG using a fast Fourier transform (FFT) with a frequency resolution of 0.18 Hz. We then determined the NREM period during light stimulation and baseline (2 min immediately before light onset), computed the mean spectrum for stimulation and baseline period for each recording, and then averaged the result for each mouse. To avoid using data during the brain state transition period, we occluded the first 10 s and the last 10 s for each NREM bout. The average EEG spectrum for each mouse was then normalized using the mean value of the EEG spectrum between 0.5 and 18 Hz (18 Hz was chosen as the upper boundary because of the stimulation-induced artifact in 20 Hz in mice expressing ChrimsonR).

**Brain state transition analysis.** Brain state transition analysis was performed according to procures described previously[25,80]. Briefly, for a given time point (5-s bin), we first determined the number of trials (x) that the mice were in a specific brain state, and then identified how many (y) of the x trails were transiting into each brain state in the next time point. The transition probability for each pair of brain states was then computed (y/x). In Fig. 7e, i and Supplementary Fig. 21, each bar represents the transition probability averaged across 12 consecutive bins. The baseline transition probabilities were averaged across all time bins within 300 s before laser onset.

## Statistical tests

**Statistic procedures.** We performed a normality test on each dataset using the Shapiro-Wilk test. The parametric tests (two-tailed paired or unpaired Student's t-tests) were used if the dataset was normally distributed ($P < 0.05$), otherwise, non-parametric tests (Wilcoxon signed-rank test or Wilcoxon rank-sum test) were used. For multiple group comparison, we used one-way or two-way ANOVA with post hoc Tukey's test. All the statistical tests were two-tailed and performed in MATLAB (2019b) and OriginLab (OriginLab Corp., 2019b). The significance level was set at $P = 0.05$. For statistics for the optogenetic experiment, we used bootstrap procedures:[80] For each experimental group with $n$ mice ($mi$ trials for mouse $i$), we resampled the data by randomly drawing $mi$ trials for each mouse (random sampling with replacement) and calculated the mean across the $n$ mice. The resampling was repeated for 100,000 iterations to get the final distribution. The confidence intervals were then extracted from the distribution of the resampled mean values. To test whether laser stimulation significantly modulated each brain state, we calculated the difference between the mean of the laser period and baseline for each bootstrap iteration. We then calculated a p-value from the resulting distribution.

**Sample size.** We did not perform a calculation on the sample size. We used a sample size comparable to studies using similar techniques and animal models[25,32,80,81].

**Data exclusion criteria.** Mice were excluded based on post hoc verification of the virus expression. No mouse was excluded for analysis.

The investigators were not blinded to the genotypes or the experimental conditions of the animals.

**Figure preparation.** All figures were prepared using Adobe Illustrator (Adobe, CS6). Diagrams in Figs. 1a, 4g, and 7a were adapted using images from biorender.com (agreement #: CS24J97GPX).

**Reporting summary**

Further information on research design is available in the Nature Portfolio Reporting Summary linked to this article.

## Data availability

The data supporting the findings of this study are included in the figures and supporting files. The raw data are available from the corresponding author upon request. Source data are provided with this paper.

## Code availability

Custom codes are available on GitHub (https://github.com/xulabsleep/MesoscaleImagingInSleep).

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

## Acknowledgements

We thank Drs. M. Poo and Z. Liang for comments or suggestions, Z.Jin and Z.Deng for technical assistance. This work was supported by the 'Strategic Priority Research Program' of the Chinese Academy of Sciences (XDB32010000 to M.X.), grants from NSFC (32221003 and 31871074 to M.X., 32170993 to S.Z.), CAS Project for Young Scientists in Basic Research (YSBR-071 to M.X.), Shanghai Municipal Science and Technology Major Project (2018SHZDZX05 and 2021SHZDZX to M.X.), and National Science and Technology Innovation 2030 Major Program (2021ZD0203704 and 2021ZD0202804 to S.Z.).

## Author contributions

M.X. conceived and supervised the projects; Yan.W. performed all transparent skull surgery with help from Z.W. and X.L.; Y.H., X.L., and X.F. performed the mesoscale imaging; Z.W. performed the optogenetic experiments; M.X., X.F., Z.W., Y.H., X.L., Yan.W., S.Z., and W.P. analyzed the data; All authors contributed to data interpretation; M.X., S.Z., and Yin.W. wrote the manuscript with inputs from all other authors.

## Competing interests

The authors declare no competing interests.
