## [Peer Review File · Nature Communications]

REVIEWER COMMENTS

Reviewer #1 (Remarks to the Author):

This is an interesting paper that provides further evidence in support of the active role of the cortex in sleep control. In my comments below, I highlight a number of important omissions, and in some cases clarification or further analysis is required. I group my comments into conceptual ones, concerning the use of terminology or reference to earlier literature, and methodological issues.

Conceptual points

1. The title refers to "sleep state switching", while the main focus of the study is on REM sleep, rather than state switching in general.
2. The authors repeatedly use "sleep regulation" while, in my opinion, they meant "sleep control". The former terminology traditionally used when describing homeostatic and circadian regulation of sleep, while control is more appropriate for state switching.
3. Two recent studies explicitly demonstrated the role of the cortex in sleep regulation and/or control, of which one is not mentioned and another is cited in passing towards the end only: (<https://www.biorxiv.org/content/10.1101/2020.07.01.179671v1> and <https://www.nature.com/articles/s41593-021-00894-6>)
4. What is meant by "global" in the title and throughout the manuscript? Please define.
5. Changes in activity obtained with imaging is difficult to interpret in relation to electrophysiologically recorded oscillations. Therefore, I suggest clarifying what is meant by "cortical activity"
6. I would recommend avoiding strong statements, such as "We have shown that the occipital cortex was highly active during REM sleep". It is relative activity, and there is a modest if any relationship of activity recorded with imaging with electrophysiological readouts.

Methodological points

1. "All experiments were performed during the daytime between 10 AM and 6 PM, and each test session lasted for 3 hours." This information is not meaningful unless it is specified at what time lights go on and off. Was it ensured that the animals were well entrained to the light-dark cycle?
2. The pictures provided show that eyes are open during sleep in head-fixed mice in your study. Did you measure the levels of light in the room where the experiments were undertaken? It is mentioned that "the ongoing neural activity was largely spontaneous", but arguably it is affected by light. I wondered if depending on the behavioural state, and, for example, pupil diameter, the visual cortex receives a varying amount of visual input, which drives the changes in cortical activity. Would the results be the same if the animals were sleeping in total darkness?
3. It is mentioned that "we normalized activity in each brain state across the entire dorsal cortex and determined the relative activation of each cortical area". I wondered if normalisation can confound the results. For example, increased relative activation in the retrosplenial or visual cortex, may instead reflect a general decrease in activity in more anterior brain areas (thus affecting the reference valued), while there is no absolute change in visual cortex and the retrosplenial cortex.
4. Both male and female mice (>8 weeks at the time of surgery) were used. Please specify how many females were used, and what was the average age.
5. It is stated that the duration of the restraint started from 10 min and gradually increased to 3 – 4 hours until they could reach stable sleep. How did you test whether it is "stable"? What was the reference point? Please provide more data on how much sleep the animals obtained while head-restrained.
6. The EEG and EMG signals were high-pass filtered at 0.5 Hz and digitized at 1 kHz. Did you use antialiasing filters?
7. It is stated that "The global signal in the imaging data, which is signal fluctuations common to the whole brain has been reported to associate with non-neuronal physiological artifacts, such as respiration and hemodynamics". However, respiration can provide an important physiological component to brain activity, and there is an abundant evidence for that (e.g. <https://pubmed.ncbi.nlm.nih.gov/35075139/>). Furthermore, arguably, the influence of respiration

etc may be locally modulated and therefore the "global signal removal" process may instead introduce artefacts.

8. It is stated "To avoid using data during the brain state transition period, we occluded the first 10 s and the last 10 s for each NREM bout. ". Please define NREM bout. I was also surprised that the last 10 sec of each NREM bout was removed (presumably including those terminating in REM sleep), which is the most interesting time for a study which investigates sleep state switching.

9. It is stated "The average EEG spectrum for each mouse was then normalized using the mean value of the EEG spectrum between 0.5 – 18 Hz". Not clear. Was it done within a state? What was the purpose of normalisation?

10. I wondered if the same population of neurons was recorded in different cortical areas?

Is it possible that different layers contributed differently to the signal in different cortical areas due to curvature of the cortex or differences in cortical thickness?

11. I found the PCA presentation a bit confusing. It is stated that PC1 – PC5 together accounted for $17.7 \pm 3.0\%$ of the total variance, but then later it says "PCA showed that REM activity was dominated by the first PC of REM sleep (PC1REM), which accounted for $27.5 \pm 2.6\%$ of the total variance".

12. Tonic vs. phasic REM was mentioned but not clear how they were defined in this study.

13. It is an intriguing observation that "cortical activity patterns during NREM oscillate between a REM-like state and an opposing state (REM-opponent state)". Can you address the influence of brief awakenings, or the relation with the infraslow oscillation, as described by Lecci et al. and others?

14. The observation that "EEG spectrum (0.5 - 20 Hz) for REM and NREM sleep during the stimulation period both had no significant difference from that during the no-stimulation period" is surprising and unexpected. Wouldn't you expect that manipulating activity of cortical region you believe is implicated in state control, directly affects brain activity during the state, which is defined by brain activity?

15. Figure 7B shows that NREM is increased during stimulation and REM is decreased, when you inhibit the occipital cortex. Firstly, I wondered whether this analysis done blindly? Secondly, I note that unfortunately the levels of NREM sleep are somewhat higher pre-stim in mCherry animals, compared to ChrimsonR. Was the 'baseline' level of sleep different between groups? I wondered if ChrimsonR and mCherry conditions need to be better balanced. Finally, I recommend using two-way ANOVA. Figure 7c shows that in one case the effect of stimulation was significant and in the other it was not, but this could be due to the difference between control groups and not stimulation. At any rate the question then arises whether stimulation is NREM-promoting rather than REM-suppressing?

Reviewer #2 (Remarks to the Author):

In Wang et al. authors investigate spontaneous global cortical activity using wide-field calcium imaging in mice to determine sleep-dependent spatiotemporal patterns of cortical activity. The authors claim that occipital activity patterns have an active role in regulating sleep states, specifically in the transition from NREM to REM sleep. The study design is simple, and the methodology used elegant. The authors perform an interesting investigation, but more details on the analysis performed would be beneficial. This work could benefit from discussing work previously published on calcium waves and functional connectivity (doi.org/10.1117/1.NPh.6.3.035002).

Introduction:

Line 59: both drosophila and spiders now also have been suggested to have sleep stages

Results:

More information is needed in this section on how different sleep states were determined (REM, NREM, wake). The authors describe this briefly in the methods section of the paper (brain state scoring 753-760) but this is not made clear to the reader in the main section of the paper.

It would be interesting to see what the brain activity looks like whilst the mouse is awake, yet not

moving. This may be a better comparison than wake while the mouse is moving.

Line 94: Were the mice able to move? Authors need to determine whether mouse activity corresponds to activity of M1 during wake

Line 132: PCs don't appear to be confined to one anatomical position in Figure 3. Could the authors please quantify the PC score across multiple regions to have a readout of these findings?

Line 148: why is the first sentence referenced?

Line 190-191: The fact that NREM changes between what the authors call a "REM-like state" and a "REM opponent state" at a 1:1.1 ratio is very interesting. This ratio could use a bit of further explanation. They state later and show in Figure 6 that this occurrence of the 'REM-like state' typically occurs just before the transition into REM sleep – it may be good then to split the earlier ratio into bouts exclusively of NREM, and into 'pre-REM' bouts (you would expect the 1:1 ration in pre-REM but not in NREM).

Line 229: Did mice induce a REM sleep debt following optogenetic silencing? Previous results suggest that the occipital cortex activity signifies switching from NREM to REM, suggesting it acts as a REM homeostat. As such, silencing this region and lowering REM sleep should induce REM increases following cessation of the stimulus

Figure 7 (and associated explanation) – it would be helpful if the authors explained further why the regions that they inhibited/activated were not the same. For the activation, only the RSP was activated, but in inhibition, it was a larger section of the occipital cortex.

Line 316: authors say that the occipital activity is acting like PGO wave; please describe why this is the case in more details.

Line 326: it would be interesting to classify the data in 'quiet wake' vs 'active wake' bouts and comment on whether RSP activity is higher during quiet wakefulness, to further support this claim. Overall: Was data in left vs right hemispheres averaged together? How is it determined whether the traces of these are the same or not? It would be beneficial if the authors could elaborate on this

It would be beneficial for the readers if the authors could better describe the definition of terms such as PCA Coefficient to make clear what an increase in the coefficient of a component implies.

Methods:

Extra detail regarding how EEG and EMG were recorded would be helpful.

Line 535: Size/make of screws needs to be listed, cortical coordinates of screw implants need to be specified.

Line 772: The brain state transition analysis could use a bit more clarification and description of the analysis. Also, what is the rationale for averaging transition probabilities?

Figures:

Figure 1:

- Scale bar for B
- Scale bar for D

Figure 2:

- Need hierarchical cluster of wake as a comparison
- Scale bars too small

Figure 3:

- Please compare the structure of PC 1-5 across multiple animals
- Please label behavioural states (top of B) clearer

Figure 4:

- B: Highlight behavioural states clearer
- C: Please plot the temporal representation of PC1 over time across animals
- D: Axis labels/scale

Figure 6: Schematics in f and h do not help convey the idea

Figure 7: Not sure about the n of this dataset

Extended F2: Please label the states that each PC/mouse represents

We thank the reviewers for the constructive comments that improved the manuscript. We have performed additional experiments and analyses, and addressed all the reviewer's concerns. We believe the revised manuscript is much improved and ready for your re-evaluation.

Reviewer #1 (Remarks to the Author):

This is an interesting paper that provides further evidence in support of the active role of the cortex in sleep control. In my comments below, I highlight a number of important omissions, and in some cases clarification or further analysis is required. I group my comments into conceptual ones, concerning the use of terminology or reference to earlier literature, and methodological issues.

Re: We thank the reviewer for the positive comments. We have performed additional experiments and analyses, and addressed the reviewer's specific concerns in the following point-to-point answers.

Conceptual points

1. The title refers to "sleep state switching", while the main focus of the study is on REM sleep, rather than state switching in general.

Re: We thank the reviewer's for the suggestion. We have now changed the title to "Global Cortical Dynamics Reveals Active Role of Occipital Cortex in Controlling REM Sleep " to describe the current study better.

2. The authors repeatedly use "sleep regulation" while, in my opinion, they meant "sleep control". The former terminology traditionally used when describing homeostatic and circadian regulation of sleep, while control is more appropriate for state switching.

Re: We have changed "sleep regulation" to "sleep control" where applicable in the revised manuscript.

3. Two recent studies explicitly demonstrated the role of the cortex in sleep regulation and/or control, of which one is not mentioned and another is cited in passing towards the end only: (<https://www.biorxiv.org/content/10.1101/2020.07.01.179671v1> and <https://www.nature.com/articles/s41593-021-00894-6>)

Re: We apologize for not citing previous work, and we have now added citations to the two mentioned work both in the introduction and discussion (Lines 40-43 and 424-427).

“...Distinct cortical activity associated with various sleep-wake stages is often considered to be a reflection of each sleep state, serving a limited role in sleep state switching⁶⁻⁹ although emerging evidence supports an active role for the cortex as well^{10,11}...”

“...The main role of the cortex in sleep-wake regulation is thought to generate homeostatic sleep pressure in an activity-dependent manner^{10,50,64}, or to support arousal⁹ or promote sleep¹¹ through top-down feedback connections to subcortical regions...”

4. What is meant by "global" in the title and throughout the manuscript? Please define.

Re: We defined "Global activity" as the large-scale neural activity across large

regions of the brain, for example, the entire dorsal part of the cortex or the entire brain. We have now included this definition in the revised text (Lines 55-57).

“...In this study, we used "global neural activity" to denote the large-scale neural activity across large areas in the brain, for example, the entire dorsal part of the cortex or the whole brain...”

5. Changes in activity obtained with imaging is difficult to interpret in relation to electrophysiologically recorded oscillations. Therefore, I suggest clarifying what is meant by "cortical activity"

Re: We have now clarified the meaning of "cortical activity" in our study. (Lines 85-88)

"...On the other hand, activity measured with mesoscale Ca^{2+} imaging reflects the summation of Ca^{2+} signals from a large number of neurons, so it may not be equivalent to electrophysiologically recorded population spiking activity. In this study, we used "cortical activity" to specifically refer to the Ca^{2+} signals obtained with widefield imaging..."

6. I would recommend avoiding strong statements, such as "We have shown that the occipital cortex was highly active during REM sleep". It is relative activity, and there is a modest if any relationship of activity recorded with imaging with electrophysiological readouts.

Re: We agree with the reviewer that the GCaMP signals are relative, and we have

avoided using strong statements in the revised manuscript.

Methodological points

1. "All experiments were performed during the daytime between 10 AM and 6 PM, and each test session lasted for 3 hours." This information is not meaningful unless it is specified at what time lights go on and off. Was it ensured that the animals were well entrained to the light-dark cycle?

Re: We thank the reviewer for pointing out this issue. All mice used in the current study were housed in a 12/12-hr light/dark cycle with the light on at 7 AM. We have now added the light-on time in the 'Methods' section (Line 441).

As for the circadian entrainment, we observed no apparent signs of circadian disruptions in our experiments. The mice had been habituated to the recording setup for about two weeks before the imaging, and they spent the majority of time in NREM sleep (65%) during the experiment, similar to the mouse sleep-wake cycle observed across the day-night cycle (our previous data (Peng et al., 2020), and data from others (Dong et al., 2022; Ingiosi et al., 2019; Li et al., 2022; Ren et al., 2018)).

References:

Dong, H., Chen, Z.K., Guo, H., Yuan, X.S., Liu, C.W., Qu, W.M., and Huang, Z.L. (2022). Striatal neurons expressing dopamine D1 receptor promote wakefulness in mice. *Curr Biol* 32, 600-613 e604.

Ingiosi, A.M., Schoch, H., Wintler, T., Singletary, K.G., Righelli, D., Roser, L.G., Medina, E., Risso, D., Frank, M.G., and Peixoto, L. (2019). Shank3 modulates sleep and expression of circadian transcription factors. *eLife* 8.

Li, S.B., Damonte, V.M., Chen, C., Wang, G.X., Kebschull, J.M., Yamaguchi, H., Bian, W.J., Purmann, C., Pattni, R., Urban, A.E., *et al.* (2022). Hyperexcitable arousal circuits drive sleep instability during aging. *Science* 375, eabh3021.

Peng, W., Wu, Z., Song, K., Zhang, S., Li, Y., and Xu, M. (2020). Regulation of sleep homeostasis mediator adenosine by basal forebrain glutamatergic neurons. *Science* 369.

Ren, S., Wang, Y., Yue, F., Cheng, X., Dang, R., Qiao, Q., Sun, X., Li, X., Jiang, Q., Yao, J., *et al.* (2018). The paraventricular thalamus is a critical thalamic area for wakefulness. *Science* 362, 429-434.

2. The pictures provided show that eyes are open during sleep in head-fixed mice in your study. Did you measure the levels of light in the room where the experiments were undertaken? It is mentioned that "the ongoing neural activity was largely spontaneous", but arguably it is affected by light. I wondered if depending on the behavioural state, and, for example, pupil diameter, the visual cortex receives a varying amount of visual input, which drives the changes in cortical activity. Would the results be the same if the animals were sleeping in total darkness?

Re: The light level in the recording box is ~80 lux, and the light level at the face of the mice is < 10 lux because the large objective lens blocked most of the light. We have added this information in the methods section of the revised manuscript.

In the revised manuscript, we performed a new analysis by examining the correlation between pupil size and activity in the primary visual cortex (V1) during both NREM and REM sleep (Supplementary Fig. 3, also shown below).

Our results suggest that the change in pupil size did not have an apparent contribution to V1 activity, because: First, although both the pupil diameter and activity in the primary visual cortex (V1) showed oscillatory dynamics during NREM sleep, there was no positive correlation between these two signals, instead,

they tend to be negatively correlated (Pearson's $r = -0.41 \pm 0.041$, mean \pm SEM) (Supplementary Fig. 3b), with a phase difference close to zero (0.50 ± 0.85 s, mean \pm SEM) (Supplementary Fig. 3c). This negative correlation suggests that both cortical activity and pupil size reflect oscillations of brain states during NREM sleep. Second, the pupil remains mostly constricted during REM sleep (Yuzgec et al., 2018); however, V1 activity often showed a large transient increase. The two signals during REM were also negatively correlated (Pearson's $r = -0.41 \pm 0.044$, mean \pm SEM) (Supplementary Fig. 3d, e). This result support that cortical activity during sleep is mainly spontaneous, unlike during wakefulness.

We have now included these results in the revised manuscript (Lines 106-108).

“...To be noted, consistent with previous reports²⁸, the head-fixed mice in our recording typically slept with their eyes open, and the pupil size varied in different sleep states (Supplementary Fig. 3)...”

Supplementary Figure 3:

Relationship between pupil size and V1 activity during recording.

(a) An example of pupil size (Scale, 1 mm) and V1 activity (Scale, $\Delta F/F_0$, 1 z-score) during NREM sleep. (b-c) Quantification of cross-correlation between the two signals. In b, $n = 12$. In c, $n = 12$. (d-e) Same as a-b, except that analysis was performed for data during the NREM to REM transition period. In d, (Scale, 1 mm and $\Delta F/F_0$, 1 z-score). In e, $n = 9$. All black bars in the figure represent 1 mm.

As for recording in total darkness, we did not perform this experiment. We reasoned that the results would be similar to the current results because the current light level is already very low, and we did not detect significant influence

due to the change in pupil size.

3. It is mentioned that "we normalized activity in each brain state across the entire dorsal cortex and determined the relative activation of each cortical area". I wondered if normalisation can confound the results. For example, increased relative activation in the retrosplenial or visual cortex, may instead reflect a general decrease in activity in more anterior brain areas (thus affecting the reference valued), while there is no absolute change in visual cortex and the retrosplenial cortex.

Re: We normalized activity across the entire dorsal cortex to determine the relative activation of each cortical area in each brain state. We agree that the scenario described by the reviewer is possible. However, we also quantified the non-normalized activity of each brain region during the sleep-wake cycle (Supplementary Fig. 4). We found that activity in the visual cortex and the retrosplenial cortex is much higher during REM sleep.

We used normalized data only in the analysis shown in Fig. 1e.

4. Both male and female mice (>8 weeks at the time of surgery) were used. Please specify how many females were used, and what was the average age.

Re: Detail information of mice used in each experiment was described in the 'Nature Research Reporting Summary'. We have now also included this information in the 'Methods' section (Lines 443-445).

5. It is stated that the duration of the restraint started from 10 min and gradually increased to 3 – 4 hours until they could reach stable sleep. How did you test whether it is "stable"? What was the reference point? Please provide more data on how much sleep the animals obtained while head-restrained.

Re: Head-fixed mice with adequate habituation show prolonged wakefulness, fragmented NREM sleep, and much less REM sleep, compared with free-behaving mice that are recorded during the same circadian time.

We used the following criteria to determine "stable" sleep: 1) NREM percentage is similar to free-moving mice (typically around 65% during the daytime) and without apparent NREM fragmentation; 2) There is a normal percentage of REM sleep (around 10%), and existing of long REM bout; 3) Lack of high percentage of wakefulness (typically less than 30%).

In our imaging experiments, the mice spent the majority of time in sleep (NREM: 65%, Wake: 27%, REM: 8%), similar to free-moving mice. This information was reported in our original manuscript (Lines 97-99).

“...The habituated mice spent the majority of time in sleep (NREM: 65%, Wake: 27%, REM: 8%) (Supplementary Fig. 1), with diverse facial movements and occasionally gross body movements during wakefulness...”

To further characterize the sleep architecture of the mice in our experiments, we have performed a new analysis of the bout duration of each sleep-wake state. This analysis is now shown in Supplementary Fig. 1 (also shown below) of the revised manuscript.

Supplementary Figure 1:

Sleep architecture of mice during the mesoscale Ca^{2+} imaging.

Shown is the distribution of bout duration for each sleep-wake state. $n = 15$ recordings from 5 mice.

6. The EEG and EMG signals were high-pass filtered at 0.5 Hz and digitized at 1 kHz.

Did you use antialiasing filters?

Re: The EEG/EMG signals were recorded using TDT RZ5 + PZ5 or Intan RHD acquisition board, for the optogenetic experiments and the imaging experiments, respectively. Both devices have been used by multiple labs to record EEG/EMG in sleep studies (Gent et al., 2018; Stucynski et al., 2022; Zhang et al., 2019; Zhong et al., 2019).

We did not use the antialiasing filters. However, the signals are unlikely distorted during digitization, because our sampling frequency (1 or 1.5 kHz) is much higher than the EEG/EMG upper frequency used in our analyses.

References:

Gent, T.C., Bandarabadi, M., Herrera, C.G., and Adamantidis, A.R. (2018). Thalamic dual control of sleep and wakefulness. *Nat Neurosci* 21, 974-984.

Stucynski, J.A., Schott, A.L., Baik, J., Chung, S., and Weber, F. (2022). Regulation of REM sleep by inhibitory neurons in the dorsomedial medulla. *Curr Biol* 32, 37-50 e36.

Zhang, Z., Zhong, P., Hu, F., Barger, Z., Ren, Y., Ding, X., Li, S., Weber, F., Chung, S., Palmiter, R.D., *et al.* (2019). An Excitatory Circuit in the Perioculomotor Midbrain for Non-REM Sleep Control. *Cell* 177, 1293-1307 e1216.

Zhong, P., Zhang, Z., Barger, Z., Ma, C., Liu, D., Ding, X., and Dan, Y. (2019). Control of Non-REM Sleep by Midbrain Neurotensinergic Neurons. *Neuron* 104, 795-809 e796.

7. It is stated that "The global signal in the imaging data, which is signal fluctuations common to the whole brain has been reported to associate with non-neuronal physiological artifacts, such as respiration and hemodynamics". However, respiration can provide an important physiological component to brain activity, and there is an abundant evidence for that (e.g. <https://pubmed.ncbi.nlm.nih.gov/35075139/>). Furthermore, arguably, the influence of respiration etc may be locally modulated and therefore the "global signal removal" process may instead introduce artefacts.

Re: We agree with the reviewer that respiration and other physiological changes can impact neural activity, and there is a possibility that the "global signal removal" process (GSR) may introduce artifacts.

Widefield imaging is a new method that has emerged in recent years (Cardin et al., 2020; Urai et al., 2022). It can record large-scale neural activity (or global activity) from the entire dorsal part of the cortex and has been widely used in many studies (Allen et al., 2017; Chan et al., 2015; Clancy et al., 2019; Couto et al., 2021; Hattori and Komiyama, 2022; Kauvar et al., 2020; Makino et al., 2017; Musall et

al., 2019; Orsolich et al., 2021; Ren and Komiyama, 2021; Scott et al., 2018; Vesuna et al., 2020; Xiao et al., 2021).

The raw imaging data are contaminated by hemodynamic signals (Couto et al., 2021; Ren and Komiyama, 2021), thus, one preprocessing step is to perform hemodynamic corrections. In general, several methods have been used in the field:

- 1) The global signal removal method, which originates from fMRI studies. The basic assumption of this method is that the global changes that are common to all brain regions largely reflect systematic hemodynamic changes.
- 2) The PCA-ICA method, which also originates from fMRI studies. This method first performs PCA-ICA to the raw data, removes the hemodynamically contaminated ICs, and uses the uncontaminated ICs to reconstruct the signals.
- 3) Using reference signals for hemodynamic correction. In this method, extra excitation light is introduced, and commonly used reference light is the isosbestic wavelength for hemoglobin (540 nm) or GCaMP (405 nm). During data processing, the reference signal is subtracted from the raw signal.

All three methods have been used by multiple studies. We used the GSR method for the following reasons:

- 1) There is no need to introduce additional excitation light to sacrifice the frame rates or to cause heavy fluorescence bleaching and tissue damage by the 405 nm ultra-violet light.
- 2) The GSR algorithm is parameter-free, minimizing human interference in the final results.
- 3) This method has a long history in processing fMRI data, and it is a commonly used method in processing widefield imaging data (Haupt et al., 2017; Murphy et al., 2016; Xie et al., 2016).

Finally, in our experiments, The global signal in our imaging data had a high similarity with the first PC when PCA was performed before GSR was implemented, and in such conditions, PC1 showed a global activation, confirming that GSR removed global correlation in the imaging data and with minimal effect on other PCs. We have noted this in the original manuscript (Lines 587-590)

“...The global signal in our imaging data had a high similarity with the first PCs when PCA was performed before GSR was implemented, and in such conditions, PC1 showed a global activation, confirming that GSR removed global correlation in the imaging data...”

Reference:

Allen, W.E., Kauvar, I.V., Chen, M.Z., Richman, E.B., Yang, S.J., Chan, K., Gradinaru, V., Deverman, B.E., Luo, L., and Deisseroth, K. (2017). Global Representations of Goal-Directed Behavior in Distinct Cell Types of Mouse Neocortex. *Neuron* 94, 891-907 e896.

Cardin, J.A., Crair, M.C., and Higley, M.J. (2020). Mesoscopic Imaging: Shining a Wide Light on Large-Scale Neural Dynamics. *Neuron* 108, 33-43.

Chan, A.W., Mohajerani, M.H., LeDue, J.M., Wang, Y.T., and Murphy, T.H. (2015). Mesoscale infraslow spontaneous membrane potential fluctuations recapitulate high-frequency activity cortical motifs. *Nat Commun* 6, 7738.

Clancy, K.B., Orsolic, I., and Mrsic-Flogel, T.D. (2019). Locomotion-dependent remapping of distributed cortical networks. *Nat Neurosci* 22, 778-786.

Couto, J., Musall, S., Sun, X.R., Khanal, A., Gluf, S., Saxena, S., Kinsella, I., Abe, T., Cunningham, J.P., Paninski, L., *et al.* (2021). Chronic, cortex-wide imaging of specific cell populations during behavior. *Nat Protoc* 16, 3241-3263.

Hattori, R., and Komiyama, T. (2022). Longitudinal two-photon calcium imaging with ultra-large cranial window for head-fixed mice. *STAR Protoc* 3, 101343.

Haupt, D., Vanni, M.P., Bolanos, F., Mitelut, C., LeDue, J.M., and Murphy, T.H. (2017). Mesoscale brain explorer, a flexible python-based image analysis and visualization tool. *Neurophotonics* 4, 031210.

Kauvar, I.V., Machado, T.A., Yuen, E., Kochalka, J., Choi, M., Allen, W.E., Wetzstein, G., and Deisseroth, K. (2020). Cortical Observation by Synchronous Multifocal Optical Sampling Reveals Widespread Population Encoding of Actions. *Neuron* 107, 351-367 e319.

Makino, H., Ren, C., Liu, H., Kim, A.N., Kondapaneni, N., Liu, X., Kuzum, D., and Komiyama, T. (2017). Transformation of Cortex-wide Emergent Properties during Motor Learning. *Neuron* 94, 880-890 e888.

Murphy, T.H., Boyd, J.D., Bolanos, F., Vanni, M.P., Silasi, G., Haupt, D., and LeDue, J.M. (2016). High-throughput automated home-cage mesoscopic functional imaging of mouse cortex. *Nat Commun* 7, 11611.

Musall, S., Kaufman, M.T., Juavinett, A.L., Gluf, S., and Churchland, A.K. (2019). Single-trial neural dynamics are dominated by richly varied movements. *Nat Neurosci* 22, 1677-1686.

Orsolich, I., Rio, M., Mrcic-Flogel, T.D., and Znamenskiy, P. (2021). Mesoscale cortical dynamics reflect the interaction of sensory evidence and temporal expectation during perceptual decision-making. *Neuron* 109, 1861-1875 e1810.

Ren, C., and Komiyama, T. (2021). Wide-field calcium imaging of cortex-wide activity in awake, head-fixed mice. *STAR Protoc* 2, 100973.

Scott, B.B., Thiberge, S.Y., Guo, C., Tervo, D.G.R., Brody, C.D., Karpova, A.Y., and Tank, D.W. (2018). Imaging Cortical Dynamics in GCaMP Transgenic Rats with a Head-Mounted Widefield Macrocope. *Neuron* 100, 1045-1058 e1045.

Urai, A.E., Doiron, B., Leifer, A.M., and Churchland, A.K. (2022). Large-scale neural recordings call for new insights to link brain and behavior. *Nat Neurosci* 25, 11-19.

Vesuna, S., Kauvar, I.V., Richman, E., Gore, F., Oskotsky, T., Sava-Segal, C., Luo, L., Malenka, R.C., Henderson, J.M., Nuyujukian, P., *et al.* (2020). Deep posteromedial cortical rhythm in dissociation. *Nature* 586, 87-94.

Xiao, D., Forys, B.J., Vanni, M.P., and Murphy, T.H. (2021). MesoNet allows automated scaling and segmentation of mouse mesoscale cortical maps using machine learning. *Nat Commun* 12, 5992.

Xie, Y., Chan, A.W., McGirr, A., Xue, S., Xiao, D., Zeng, H., and Murphy, T.H. (2016). Resolution of High-Frequency Mesoscale Intracortical Maps Using the Genetically Encoded Glutamate Sensor iGluSnFR. *The Journal of neuroscience : the official journal of the Society for Neuroscience* 36, 1261-1272.

8. It is stated "To avoid using data during the brain state transition period, we occluded the first 10 s and the last 10 s for each NREM bout. ". Please define NREM bout. I was also surprised that the last 10 sec of each NREM bout was removed (presumably

including those terminating in REM sleep), which is the most interesting time for a study which investigates sleep state switching.

Re: We used the standard definition for NREM bout in the field, which is a short period of NREM, starting from another brain state transitioning into NREM, and ending with transitioning to another brain state.

This analysis is meant to examine whether the NREM during stimulation has a similar EEG profile to that during baseline. To examine this, we compared the EEG spectrum during the two conditions. We removed the first and last 10 s of each NREM bout because they contain transition phases, which have mixed patterns of EEG profiles.

We agree with the reviewer that the transition phases are very interesting, and we also analyze how laser stimulation affects brain state transition in our original manuscript (Fig. 7e).

9. It is stated "The average EEG spectrum for each mouse was then normalized using the mean value of the EEG spectrum between 0.5 – 18 Hz". Not clear. Was it done within a state? What was the purpose of normalisation?

Re: The purpose of this analysis was to examine whether the optogenetic stimulation-induced NREM is similar to that of natural NREM, as it is possible that manipulation-induced sleep may be an unnatural state (e.g., many drugs can induce sleep-like states with unnatural EEG patterns). The specific index we used to measure the similarity was the EEG power spectral density, which quantifies

EEG power at each frequency. We used the frequency range of 0.5 - 18 Hz because the optogenetic manipulation caused a stimulation artifact at 20 Hz. We normalized the EEG power spectrum, as the raw power can be different for each mouse (possibly due to variations in EEG electrodes and/or EEG insertion). This procedure has been used extensively in previous studies (Foley et al., 2017; Izawa et al., 2019; Ren et al., 2018).

This analysis was performed for individual brain states. We compared EEG patterns for both NREM and REM sleep (Fig. 7d).

Reference:

Foley, J., Blutstein, T., Lee, S., Erneux, C., Halassa, M.M., and Haydon, P. (2017). Astrocytic IP3/Ca(2+) Signaling Modulates Theta Rhythm and REM Sleep. *Front Neural Circuits* 11, 3.

Izawa, S., Chowdhury, S., Miyazaki, T., Mukai, Y., Ono, D., Inoue, R., Ohmura, Y., Mizoguchi, H., Kimura, K., Yoshioka, M., *et al.* (2019). REM sleep-active MCH neurons are involved in forgetting hippocampus-dependent memories. *Science* 365, 1308-1313.

Ren, S., Wang, Y., Yue, F., Cheng, X., Dang, R., Qiao, Q., Sun, X., Li, X., Jiang, Q., Yao, J., *et al.* (2018). The paraventricular thalamus is a critical thalamic area for wakefulness. *Science* 362, 429-434.

10. I wondered if the same population of neurons was recorded in different cortical areas? Is it possible that different layers contributed differently to the signal in different cortical areas due to curvature of the cortex or differences in cortical thickness?

Re: We agree with the reviewer that it is possible that signals in different cortical regions may consist of signals from different layers due to the difference in cortical thickness and curvatures. However, these factors may not have a big impact on our signals, as explained below: 1) The Thy1-GCaMP6 mice we used mainly label

a homogenous group of neurons in layer 2/3 and layer 5. 2) The objective lens is a low magnification (2x), which can collect light from a broader depth, unlike the high magnification lens. 3) Depth of the signals mainly changes in intensity, and dF/F_0 can overcome the difference in signal intensity. Finally, the functional role of the occipital cortex in controlling REM sleep has been validated by our optogenetic experiments.

11. I found the PCA presentation a bit confusing. It is stated that PC1 – PC5 together accounted for $17.7 \pm 3.0\%$ of the total variance, but then later it says "PCA showed that REM activity was dominated by the first PC of REM sleep ($PC1^{REM}$), which accounted for $27.5 \pm 2.6\%$ of the total variance".

Re: We apologize for the confusion. The PCA in Fig. 3 used data from all three brain states, whereas $PC1^{REM}$ was calculated using only data from REM sleep.

We have now added a note in the revised manuscript to clarify this (Lines 189-190).

"...PCA (using only data in REM sleep) showed that REM activity was dominated by the first PC of REM sleep ($PC1^{REM}$)..."

12. Tonic vs. phasic REM was mentioned but not clear how they were defined in this study.

Re: In the revised manuscript, we have performed a new analysis on the relation between $PC1^{REM}$ and different states of REM sleep (tonic vs. phasic). According to

the literature, we defined phasic and tonic REM as REM sleep with or without phasic muscle movements (we used facial movements extracted from the face camera). We found that the $PC1^{REM}$ was significantly higher during phasic REM (Supplementary Fig. 12, also shown below).

This analysis is now added to the revised manuscript (Lines 199-203, Supplementary Fig. 12).

“...We defined phasic and tonic REM according to the presence of phasic facial movements (extracted from the face camera) and found that the $PC1^{REM}$ was significantly higher ($P < 0.001$, Wilcoxon signed rank test) (Supplementary Fig. 12) during phasic REM, suggesting different REM stages exhibit distinct cortical activation patterns...”

Supplementary Figure 12:

$PC1^{REM}$ during tonic and phasic REM

(a) An example of facial movements (scale, 200 a.u.) and $PC1^{REM}$ (scale, 0.03). The Red line indicates the mean. (b) Quantification of $PC1^{REM}$ in tonic and phasic REM. $P < 0.001$, Wilcoxon signed-rank test. $n = 12$ sessions from 5 mice.

13. It is an intriguing observation that "cortical activity patterns during NREM oscillate between a REM-like state and an opposing state (REM-opponent state)". Can you address the influence of brief awakenings, or the relation with the infraslow oscillation, as described by Lecci et al. and others?

Re: We have performed a new analysis to examine the relation between the REM-like state and the brief awakenings (microarousal) and the infraslow oscillations. For the influence of microarousal, we detected microarousal events as the brief increase in EMG power during NREM sleep and used them to align the REM-like state (CC with PC1^{REM}). We found that CC significantly increased before microarousal and decreased after microarousal ($P < 0.001$ for both comparisons, one-way repeated measures ANOVA with posthoc Tukey's test).

This result is now shown in the revised manuscript (Lines 261-264 , Supplementary Fig. 17, also shown below).

“...We found that there were more REM-like states before microarousal events and more REM-opponent events during microarousal events ($P < 0.001$ for both comparisons, one-way repeated measures ANOVA with posthoc Tukey's test) (Supplementary Fig. 17)...”

Supplementary Figure 17:

PC1^{REM}-like activity and microarousal event

(a) Two examples showing PC1^{REM}-like activity during microarousal events (highlighted in the blue box). Scale, EMG power 100 a.u.; PC1^{REM} coefficient (CC), 0.3. (b) PC1^{REM} CC aligned to the onset of microarousal events (n = 69 from 5 mice). Scale (CC), -0.4 - 0.4. (c) Quantification of change in PC1^{REM} from all five mice. Statistical comparison was performed between the 3rd group (from -10 s to onset of microarousal events) and the other five groups. ***P < 0.001; n.s., P > 0.05 (one-way repeated measures ANOVA with Tukey's posthoc test). n = 69.

For the relation with the infra-low oscillations in the sigma band, we calculated the correlation between the two signals. We found a significant positive correlation between the two signals (Pearson's $r = 0.47 \pm 0.02$).

We have now included this analysis in the revised manuscript (Fig. 5g-j, Lines 251-257, also shown below)

"... It has been recently found that EEG during NREM sleep exhibits infra-slow oscillations in the sigma band⁴²⁻⁴³. We next examined the relationship between the sigma oscillations and the REM-like activity. We chose long NREM bouts with few

micro-arousals and calculated EEG sigma power (Fig. 5g). Cross-correlation analysis revealed a significant positive correlation (Pearson's $r = 0.47 \pm 0.02$) between the two signals (Fig. 5h, i), with no apparent difference in peak time at the group level although the sigma oscillations may lead or lag the REM-like activity for a few seconds (Fig. 5h, j)..."

Figure 5g-j:

(g) Top to bottom, EEG power spectrogram, Sigma oscillations (scale, 0.5 a.u.), coefficient of PC1^{REM} (scale, 0.02 and 100 s). (h-i) Cross-correlation analysis of EEG sigma oscillations and PC1^{REM} coefficient. $n = 15$ recordings from 5 mice.

14. The observation that "EEG spectrum (0.5 - 20 Hz) for REM and NREM sleep during the stimulation period both had no significant difference from that during the no-stimulation period" is surprising and unexpected. Wouldn't you expect that manipulating activity of cortical region you believe is implicated in state control, directly affects brain activity during the state, which is defined by brain activity?

Re: We agree with the reviewer that manipulating cortical activity would produce a difference in the sleep state as the latter is defined by brain activity. This is also

one of the points we wanted to promote in our manuscript. Indeed, our optogenetic silencing profoundly suppressed REM sleep and increased NREM sleep.

The original description meant that our optogenetic stimulation did not produce unnatural brain states, as supported by our analyses that the EEG spectrum for both NREM and REM during the stimulation period was not significantly different from those during baseline.

We have now revised the text (Lines 318-321).

“... Furthermore, optogenetic stimulation did not produced unnatural brain states, as the EEG spectrum (0.5 - 18 Hz) for both REM and NREM sleep during the stimulation period had no significant difference from that during the no-stimulation period...”

15. Figure 7B shows that NREM is increased during stimulation and REM is decreased, when you inhibit the occipital cortex. Firstly, I wondered whether this analysis done blindly? Secondly, I note that unfortunately the levels of NREM sleep are somewhat higher pre-stim in mCherry animals, compared to ChrimsonR. Was the 'baseline' level of sleep different between groups? I wondered if ChrimsonR and mCherry conditions need to be better balanced. Finally, I recommend using two-way ANOVA. Figure 7c shows that in one case the effect of stimulation was significant and in the other it was not, but this could be due to the difference between control groups and not stimulation. At any rate the question then arises whether stimulation is NREM-promoting rather than REM-suppressing?

Re: In the optogenetic stimulation experiment, we did not score the brain states

blindly because the stimulation induced a 20-Hz artifact that can be seen from the EEG spectrogram of the ChrimsonR-expressing mice during brain state scoring. We have noticed this in the manuscript (Line 760).

"...The investigators were not blinded to the genotypes or the experimental conditions of the animals..."

As for the different levels of NREM sleep during baseline, we agree that the mChreey group showed a slightly higher level of NREM than that in the ChrimsonR group. This difference was likely caused by homeostatic sleep regulation: In the ChrimsonR group, laser stimulation increased NREM sleep during the stimulation period, thus, the mice may spend less time in the following baseline periods.

For the statistical analysis, we changed the original method to two-way ANOVA with repeated measures, as suggested by the reviewer. We found that there was significant main effect of laser stimulation on the percentage of both REM and NREM sleep but not wakefulness (Wake, $F(1, 8) = 0.097$, $P = 0.76$; NREM, $F(1, 8) = 13.0$, $P = 0.007$; REM, $F(1, 8) = 25.5$, $P < 0.001$; two-way ANOVA with repeated measures). We also performed a post hoc comparison and found no significant difference in the baseline NREM level between the groups ($P = 0.28$, post hoc Tukey's test). The new result is consistent with our previous statistical analysis. We have noted this in the revised manuscript (Lines 310-314).

"... The laser stimulation-induced modulation in the ChrimsonR- and mCherry-expressing mice were also compared using two-way repeated measures ANOVA, which

revealed significant main effects of laser stimulation on the percentage of both REM and NREM sleep but not wakefulness (Wake, $F(1, 8) = 0.097$, $P = 0.76$; NREM, $F(1, 8) = 13.0$, $P = 0.007$; REM, $F(1, 8) = 25.5$, $P < 0.001$; two-way repeated measures ANOVA)...”

As for whether the optogenetic inhibition was NREM-promoting vs. REM-suppression, we prefer that the primary effect is REM suppression, as described below. First, in mice, REM and NREM are normally tightly linked when we look at the time of NREM and REM sleep in each hour across the day-night cycle—if we see more NREM in a given hour, we tend to also see more REM sleep. However, in our experiment, we found a dissociation between NREM and REM—NREM increased, but REM decreased, suggesting that the main effect is likely to be REM suppression. Second, our brain state transition analysis revealed that optogenetic inhibition had a significant effect ($P < 0.001$) on reducing NREM→REM transition and increasing NREM consolidation (NREM→NREM transition), suggesting that the mice had difficulty entering REM sleep. Therefore, we believe that the main effect of optogenetic silencing is suppressing REM.

Reviewer #2 (Remarks to the Author):

In Wang et al. authors investigate spontaneous global cortical activity using widefield calcium imaging in mice to determine sleep-dependent spatiotemporal patterns of cortical activity. The authors claim that occipital activity patterns have an active role in regulating sleep states, specifically in the transition from NREM to REM sleep. The study design is simple, and the methodology used elegant. The authors perform an interesting investigation, but more details on the analysis performed would be beneficial. This work could benefit from discussing work previously published on calcium waves and functional connectivity (doi.org/10.1117/1.NPh.6.3.035002).

Re: We thank the reviewer for the positive comments. We have performed additional experiments and analyses, and addressed the reviewer's specific concerns in the following point-to-point answers.

We apologize for not citing previous work. The mentioned paper used mesoscale Ca²⁺ imaging to record cortical activity during the sleep-wake cycle and under anesthesia. The authors found a large change in GCaMP delta oscillation in different states, while the high-frequency components were preserved across states. In general, the slow oscillation in delta bands and the activity patterns revealed using seeds-based correlation analysis are consistent with our results (Fig. 3a). We have cited the work accordingly in the revised manuscript (Lines 159-160).

“...PCA reliably extracted highly similar activity patterns from different mice (Supplementary Fig. 7), similar to previous analysis using seeds-based correlation...”

Introduction:

Line 59: both drosophila and spiders now also have been suggested to have sleep stages

Re: We have added this literature in the revised manuscript (Line 40).

“...The different sleep stages are also observed in non-mammalian species...”

Results:

1) More information is needed in this section on how different sleep states were determined (REM, NREM, wake). The authors describe this briefly in the methods section of the paper (brain state scoring 753-760) but this is not made clear to the reader in the main section of the paper.

Re: We have now described the brain state scoring procedures in the main section (Lines 92-97).

"...EEG (from the left auditory cortex) and EMG (from the neck muscle) were recorded to determine the sleep-wake states of the mice. To score the brain states, we performed fast Fourier transforms (FFTs) on EEG to extract δ (0.5 - 4 Hz) and θ (6 - 10 Hz) activity. Brain states were classified according to established criteria: Wakefulness, desynchronized EEG and high EMG activity; NREM, synchronized EEG with high δ activity and low EMG activity; REM, high EEG θ power and low EMG activity..."

2) It would be interesting to see what the brain activity looks like whilst the mouse is awake, yet not moving. This may be a better comparison than wake while the mouse is moving.

Re: We have performed a new analysis to examine cortical activity during active and quiet wakefulness. Unlike traditional detection of active wakefulness, which is often defined as an awake period with gross body movement, we can only extract facial movements from our data and define different phases of wakefulness accordingly. The analysis showed that the two types of wakefulness are associated with different cortical activation patterns.

This result is now included in the revised manuscript (Lines 128-129, Supplementary Fig. 5, also shown below).

“...Additionally, active wakefulness (with more facial movements) and quiet wakefulness were also associated with different cortical patterns...”

Supplementary Figure 5:

Normalized activity in each brain region during active and quiet wakefulness

(a) Average Ca²⁺ activity during active and quiet wakefulness. Scale ($\Delta F/F_0$, z-score): -1.2 - 1.2. The black bar is 1 mm. (b) Normalized activation across the recorded brain regions in active and quiet wakefulness. n = 15 sessions from 5 mice.

3) Line 94: Were the mice able to move? Authors need to determine whether mouse activity corresponds to activity of M1 during wake

Re: The mice were able to move their body and paws freely. We have performed a new analysis to examine the relation between mouse activity and cortical activity recorded in our widefield imaging.

During imaging, we captured the mouse face using an infra camera, and we can only extract eye and facial movements from the video due to the limited cover of the camera (we need a large magnification lens to capture the pupil; thus, the camera only covered the face of the mouse). We used the open-source software "FaceMap software (www.github.com/MouseLand/FaceMap)" to extract the facial movements. We then calculated the correlation between facial movements and cortical Ca²⁺ signals to get a correlation map. The analysis reveals that facial movements have a high correlation with signals in the somatosensory cortex and the motor cortex (Supplementary Fig. 2, also shown below).

These results are now included in the revised manuscript (Lines 102-104, Supplementary Fig. 2).

“...we found that the facial movements (extracted from the face camera) were significantly associated with elevated Ca²⁺ activity in the somatosensory cortex and

motor cortex during wakefulness...”

Supplementary Figure 2:

Cortical activity associated with facial movements

Shown is the mean correlation map between cortical activity and facial movements. $n = 9$ recordings from 5 mice. Scale, CC -0.3-0.3. Black bar, 1 mm.

4) Line 132: PCs don't appear to be confined to one anatomical position in Figure 3. Could the authors please quantify the PC score across multiple regions to have a readout of these findings?

Re: The quantification of PC scores across brain regions was shown in the supplementary figure of the original manuscript (Supplementary Fig. 8 of the revised manuscript).

5) Line 148: why is the first sentence referenced? The PCs exhibited prominent oscillations^{23,24}

Re: We have now rewritten the sentence (Line 182).

"... The PCs exhibited prominent oscillations, similar to previous reports³⁰⁻³²..."

6) Line 190-191: The fact that NREM changes between what the authors call a "REM-like state" and a "REM opponent state" at a 1:1.1 ratio is very interesting. This ratio could use a bit of further explanation. They state later and show in Figure 6 that this occurrence of the "REM-like state" typically occurs just before the transition into REM sleep – it may be good then to split the earlier ratio into bouts exclusively of NREM, and into 'pre-REM' bouts (you would expect the 1:1 ration in pre-REM but not in NREM).

Re: We thank the reviewer for the suggestion. We agree with the reviewer that the ratio in pre-REM and NREM should be different.

The proposed analysis is equivalent to our analysis in Fig. 6h-i, in which we calculated the percentage of REM-like states during different phases of NREM and pre-REM. The amount of REM-like states is proportional to the ratio between the "REM-like state" and "REM opponent state". We can see that pre-REM has much more REM-like states (Fig. 6i), supporting a higher ratio than NREM.

7) Line 229: Did mice induce a REM sleep debt following optogenetic silencing? Previous results suggest that the occipital cortex activity signifies switching from NREM to REM, suggesting it acts as a REM homeostat. As such, silencing this region and lowering REM sleep should induce REM increases following cessation of the stimulus

Re: In the optogenetic silencing experiment, the probability of REM sleep immediately after the stimulation is higher than that before stimulation, indicating a build-up of REM debt during the optogenetic silencing period.

To further validate this finding, we performed a new experiment to silence the occipital cortex for a longer time (20 min), and we also used stronger laser power (20 mW total, twice the power used in our original optogenetic silencing experiment). We found that prolonged optogenetic silencing efficiently suppressed REM sleep throughout the laser period, and mice often entered REM sleep shortly after we turned off the laser. This result strongly suggested that optogenetic silencing of the occipital cortex produced REM sleep debt. We have now incorporated this experiment into the revised manuscript (Lines 353-368 and Fig. 7, also shown below).

“...Taken together, we have demonstrated that occipital activity signifies REM propensity and plays an active role in controlling REM sleep. An immediate intriguing question is whether inhibition of occipital activity leads to an increase in REM pressure since REM sleep is also homeostatically regulated^{46,47}. In our optogenetic inhibition experiment, we did observe REM rebound shortly after the laser stimulation (Fig. 7b, Supplementary Fig. 19), suggesting the possibility of increased REM pressure by suppressing occipital activity. To further examine this, we performed new experiments in which we inhibited occipital activity for a longer time (Laser on, 20 minutes; Laser off, 5-8 minutes) and with higher laser power (20 mW, to achieve more significant inhibition) (Fig. 7j). We found that prolonged inhibition of occipital activity effectively

suppressed REM sleep throughout the stimulation period—the percentage of REM sleep decreased to 3.5% compared with 10.5% before stimulation ($P = 0.032$, one-way ANOVA with posthoc Tukey's test) (Fig. 7j-l). Importantly, mice frequently entered into REM shortly after stimulation, with a substantial increase in the percentage of REM sleep (25.9%; $P < 0.001$, one-way ANOVA with posthoc Tukey's test, compared with baseline) (Fig. 7j-l). These results show that inhibition of occipital activity produces REM sleep pressure, suggesting that the occipital cortex plays a role in REM homeostasis...”

Figure 7j-l:

(j) Schematic of the experiment showing prolonged optogenetic suppression of the occipital cortex (up) and an example recording (scale, 0.5 mv and 5 minutes). (k) Percentage of time in each brain state during light stimulation (yellow) in mice expressing ChrimsonR. $n = 30$ sessions from 6 mice (5 sessions/mouse). Shading, s.e.m. (l) Histogram showing the percentage of time in REM sleep before (gray), during (yellow), and after (magenta). Before vs. Stimuli., $P = 0.032$; After vs. Stimuli., $P < 0.0002$; Before vs. After., $P < 0.0001$; One-way repeated measures ANOVA with Tukey's posthoc test.

8) Figure 7 (and associated explanation) – it would be helpful if the authors explained further why the regions that they inhibited/activated were not the same. For the activation, only the RSP was activated, but in inhibition, it was a larger section of the occipital cortex.

Re: We inhibited a larger cortical region because our imaging experiment showed that these regions had strong activation during REM sleep. On the other hand, we only activated part of the RSP because optogenetic activation of large areas may lead to a broad activation and cause a risk of seizure. We have made a note of this in the manuscript (Lines 343-345).

“... Since excessive activation of cortical excitatory neurons can cause seizure, we only expressed ChrimsonR (driven by CaMKII promotor) in a small region of the occipital cortex, the RSP...”

Another reason the inhibited region is larger than the activated region is that loss-of-function experiments often require relatively complete removal of the contributing factors in order to have a prominent effect on the behavior level, while gain-of-function experiments may only need to activate one of the contributing factors.

For these two sets of experiments, we would like to say that the inhibition experiment certainly carried more weight since it tells us the physiological role of the suppressed brain regions.

9) Line 316: authors say that the occipital activity is acting like PGO wave; please describe why this is the case in more details.

Re: We suspected that the occipital activity might represent the cortical feature of the PGO waves in mice. Because 1) Both occipital activity and PGO waves are associated with REM sleep; 2) There is increased activity in the occipital cortex in both cases. To further test this idea, we performed a new experiment in which we imaged cortical activity and recorded the pontine LFP simultaneously.

We identified the P-wave (pontine part of the PGO waves) in the pontine LFP according to previous work and examined how the P-wave related to the occipital activity. Our results showed that there was a high correlation between the two signals, suggesting that the occipital activity is likely to reflect the occipital activation that is equivalent to PGO waves in cats and primates. These new results are now shown in Fig. 4g-i of the revised manuscript (Lines 216-224, also shown below).

“... Another feature of brain activity during REM sleep is the PGO waves, which originate in the pons and propagate through the lateral geniculate nucleus to the occipital cortex³⁹. The unique occipital activity we observed may represent the cortical activity of mouse PGO waves. We therefore examined this by recording local field potentials (LFP) from the pons during the mesoscale imaging (Fig. 4g). We identified the pontine waves (P-wave, the pontine part of the PGO waves) as large negative potential in the LFP^{40,41}, and we found that occipital activity significantly increased immediately after the P-wave (Fig. 4h, i; peak latency, 0.89 ± 0.07 s, mean \pm SEM).

This result support that mice also have PGO waves, and the structured occipital activity during REM sleep may represent the cortical activation of the PGO waves...”

Figure 4g-i:

(g) Left, schematic diagram depicting LFP recording from the pons. Right, Examples of P-waves (Scale, 0.1 mV and 1s) and associated cortical activity (Scale, $\Delta F/F_0$, z-score: -1.5 - 1.5). (h-i) Occipital activity increased immediately after the P-waves. PC1^{REM} coefficient aligned to the negative peak of p-waves detected from one recording (h, n = 33) or from all 5 mice (i, n = 199). Scale in h, $\Delta F/F_0$ (z-score): -1.5 - 1.5. All black bars in the figure represent 1 mm.

10) Line 326: it would be interesting to classify the data in 'quiet wake' vs 'active wake' bouts and comment on whether RSP activity is higher during quiet wakefulness, to further support this claim.

Re: We thank the reviewer for the suggestions. In our analysis of cortical activity patterns during active and quiet wakefulness (states defined using facial movements) (Supplementary Fig. 5), we found that RSP activity is not as high during quiet wakefulness as that during REM sleep. A possible explanation is that the 'quiet wakefulness' states we defined still contain multiple behavior states. We

thus examined the raw trace during wakefulness and found that the RSP showed increased activity often when the mice were in drowsy states (the pupil was constricted and lack of facial movements) (see example traces from three different mice) (Supplementary Fig. 15, also shown below).

This result is now included in the revised manuscript (Line 233).

“... However, while cortical patterns during wakefulness were generally not similar with the PC1^{REM} (except during the drowsy states when the pupil was constricted and lack of facial movements. Supplementary Fig. 15)...”

Supplementary Figure 5:

Normalized activity in each brain region during active and quiet wakefulness

(a) Average Ca^{2+} activity during active and quiet wakefulness. Scale ($\Delta F/F_0$, z-score):

-1.2 - 1.2. The black bar is 1 mm. (b) Normalized activation across the recorded brain regions in active and quiet wakefulness. n = 15 sessions from 5 mice.

Supplementary Figure 15:

PC1^{REM}-like activity during drowsy states

Shown are three examples of PC1^{REM}-like activity during drowsy states. Drowsy states (highlighted in the blue box) were manually identified when the pupil was constricted and lack of facial movements. Scale, CC, 0.5; pupil size, 0.3 mm; facial movement, 1 a.u.

11) Overall: Was data in left vs right hemispheres averaged together? How is it determined whether the traces of these are the same or not? It would be beneficial if the authors could elaborate on this

Re: Activity from the left vs. right hemispheres were averaged together because we found that the majority of cortical activity in our recording seems to be symmetric. In the PCs, we can see that all major PCs (PC1-5) are symmetric (Supplementary Fig. 8b); however, many of the minor PCs are asymmetric (Supplementary Fig. 7), suggesting that large dynamic changes in the dorsal cortex are highly similar across the two hemispheres, but some of the events are unique to one hemisphere.

To quantify the symmetric nature of the major PCs, we calculated the correlation between the left and right parts of PC1-5. The result shows high correlations for all the PCs (mean CC: 0.94, 0.93, 0.92, 0.80, and 0.90 for PC1-5, respectively). This

result is now included in Supplementary Fig. 8b of the revised manuscript (also shown below).

Supplementary Figure 7a:

(a) The first ten PCs in each mouse

Supplementary Figure 8b:

(b) Cross-correlation between PC1-5 in the two hemispheres. Data were from 5 mice.

12) It would be beneficial for the readers if the authors could better describe the definition of terms such as PCA Coefficient to make clear what an increase in the coefficient of a component implies.

Re: We thank the reviewer for the suggestion. We have added a description for the PCA Coefficient in the revised manuscript (Lines 170-172).

"... These PCs exhibited distinct dynamics during sleep-wake cycles (Fig. 3b), which were reflected in changes in PCA coefficients, a quantitative measure of the correlation between each PC and cortical activity patterns during each imaging frame—higher coefficients mean higher similarity ..."

Methods:

1) Extra detail regarding how EEG and EMG were recorded would be helpful.

Re: We have now added details for EEG/EMG recording in the Methods section (Lines 521-533, also shown below).

"... For polysomnography recordings during widefield imaging, the EEG and EMG signals (high-pass filtered at 0.5 Hz and digitized at 1 kHz) were acquired using an RHD acquisition board from Intan Technologies. The acquisition board was also used to record the exposure-out signal from the camera to synchronize the EEG/EMG signals with the imaging data.

For polysomnography recordings in optogenetic manipulation experiments, mice were transferred into recording cages placed in a sound-attenuation box (80 x 100 x 120 cm) and connected to the amplifier (TDT system-3 amplifier RZ2 + PZ5) with a flexible recording cable via a commutator. EEG/EMG were high-pass filtered at 0.5 Hz and digitized at 1526 Hz. Mice were habituated for two days before recording. During

recording, a large power LED (for optogenetic silencing experiments) or optical fibers (for optogenetic activation experiments) were attached to the head implant. The recording session started after 30 min and lasted for three hours..."

2) Line 535: Size/make of screws needs to be listed, cortical coordinates of screw implants need to be specified.

Re: We apologize for missing this information. We have now added this information in the "Methods" section of the revised manuscript (Lines 455-458, also shown below).

"... Two stainless steel screws (M1.0 x 3 mm) were inserted into the skull above the auditory cortex (AP -3.5 mm, ML 4.0 mm) of both hemispheres, and another two screws were inserted into the skull above the olfactory bulb (AP 3.7 mm, ML 0.8 mm) and the cerebellum (AP -5.5 mm, ML 1.0 mm), respectively...."

3) Line 772: The brain state transition analysis could use a bit more clarification and description of the analysis. Also, what is the rationale for averaging transition probabilities?

Re: The brain state transition analysis was the same as that used previously. We have clarified the brain state transition analysis in the revised manuscript (Lines 324-330, also shown below).

"... In the optogenetic silencing experiments, we observed stimulus-induced reductions in REM sleep, which may be the result of reduced REM entry or reduced REM

maintenance. Brain state transition analysis can distinguish these possibilities. For a given time point (5-s bin), brain state transition analysis determines the probability of the current brain state transitioning to other brain states. In the analysis, we compared the transition probability during the baseline and stimulation periods (Fig. 7e, f, Supplementary Fig. 21). Each bar in the figure represents the transition probability averaged in one minute. ..."

The brain state transition analysis was performed for each 5-s bin during both baseline and laser periods, while in the bar plot, each bar represents the mean transition probability for each pair of brain states in one minute (12 bins). Thus, averaging was used to quantify the mean probability for each transition during a specific period, and the averaging or the number of averaged bins does not significantly change the statistical results.

Figures:

1) Figure 1:

- Scale bar for B- Scale bar for D

Re: Added

2) Figure 2:

- Need hierarchical cluster of wake as a comparison

- Scale bars too small

Re: We have added hierarchical clustering for wakefulness and re-arranged this

figure to enlarge the scale.

3) Figure 3:

- Please compare the structure of PC 1-5 across multiple animals

Re: We have added a new analysis to compare the structure of PC 1-5 across animals. The analysis revealed highly similarity of all five PCs (mean CC: 0.90, 0.85, 0.76, 0.65, and 0.76 for PC1-5, respectively). This result is now included in the Supplementary Figure 7b, c of the revised manuscript (also shown below).

Supplementary Figure 7b-c:

(b) Cross-correlation of PC1-5 across mice. Note that PC4 and PC5 were swapped during analysis. (c) Quantification of pairwise cross-correlation in b. Data were from 5 mice.

- Please label behavioural states (top of B) clearer

Re: The behavioral states in the figure are relabeled.

4) Figure 4:

- B: Highlight behavioural states clearer

Re: The behavioral states in the figure are relabeled.

- C: Please plot the temporal representation of PC1 over time across animals (new suppl figure)

Re: We have included two additional examples from another two mice showing the PC1^{REM} dynamics during REM sleep in Supplementary Figure 11 (also shown below) of the revised manuscript.

Supplementary Figure 11:

Dynamic changes in PC1^{REM} from two other mice

Shown are two additional examples of PC1^{REM} from two other mice. Scale, 0.03.

- D: Axis labels/scale

Re: Added.

5) Figure 6: Schematics in f and h do not help convey the idea

Re: We have made new schematics in the revised manuscript.

6) Figure 7: Not sure about the n of this dataset (in legends)

Re: The number of mice used in each experiment was in the figure legends and 'Nature Research Reporting Summary'. We have now added this information also in the Figure (Fig. 7).

Note that, for the RSP activation experiment, the original data were from 4 mice. We have repeated this experiment in two more mice, so this dataset now has 6 mice.

7) Extended F2: Please label the states that each PC/mouse represents

Re: Added

REVIEWERS' COMMENTS

Reviewer #1 (Remarks to the Author):

Thank you for the effort to address my comments.

Reviewer #2 (Remarks to the Author):

The authors have addressed my concerns.